# Dynamic and transformable Cu$_{12}$ cluster-based C-H···π-stacked porous supramolecular frameworks

Chengkai Zhang[1], Zhi Wang [1], Wei-Dan Si[1], Hongxu Chu[1], Lan Zhou[1], Tong Li[1], Xian-Qiang Huang[2], Zhi-Yong Gao[3], Mohammad Azam[4], Chen-Ho Tung[1], Ping Cui[1] & Di Sun [1] ✉

The assembly of cluster-based π-stacked porous supramolecular frameworks presents daunting challenges, including the design of suitable cluster building units, control of the sufficient C-H···π interactions, trade-off between structural dynamics and stability as well as understanding the resulting collective properties. Herein, we report a cluster-based C-H···π interaction-stacked porous supramolecular framework, namely, Cu12a-π, consisting of Cu$_{12}$ nanocluster as a 6-connected node, which is further propagated to a dynamic porous supramolecular frameworks via dense intralayer C-H···π interactions, yielding permanent porosity. In addition, Cu12a-π can be transformed into cluster-based nonporous adaptive crystals (Cu12b-NACs) via ligand-exchange following a dissociation-reassembly mechanism. Moreover, Cu12a-π can efficiently remove 97.2% of iodine from saturated iodine aqueous solutions with a high uptake capacity of 2.96 g·g$^{-1}$. These prospective results positioned at cluster-based porous supramolecular framework and enlighten follow-up researchers to design and synthesize such materials with better performance.

Inspired by fascinating molecular pores that nature has evolved, scientists have made great progress in constructing porous materials[1-5]. Recently, a series of crystalline porous materials (CPMs) were assembled through intermolecular weak non-covalent interactions, which involve hydrogen bond[6-10], π···π or C-H···π stacking[11-16], electrostatic interactions, van der Waals forces[17,18], etc. Wherein, π-stacked organic frameworks (πOFs), one of the newly emerging CPMs, have greatly broadened the boundaries of porous science, but few examples associated to such CPMs have been reported. Moreover, most of the reported πOFs are stabilized not merely by pure π···π stacking interactions, whereas other forces are also involved to a greater or lesser extent, e.g. C-H···π stacking interactions[12]. As a result, it is hard to figure out how much the C-H···π stacking interactions contribute the stability

of CPMs. Therefore, the construction of supramolecular networks with permanent pores stabilized by C-H···π stacking interactions is meaningful to investigate the performance of such a non-covalent interaction in CPMs.

In general, the CPMs stabilized by C-H···π interactions may show the similar dynamic behavior like third-generation MOFs due to the characteristic of non-covalent interactions, and can reversibly transform their structures in response to chemical guests via physical stimulation[19-21]. The rational construction of such CPMs may require the following two main factors: 1) the robust building unit; 2) the medium strength of the bridge linker, where the frame structure can balance the stability and flexibility. However, the most reported πOFs or HOFs (hydrogen-bonded organic frameworks) are constructed by

[1]School of Chemistry and Chemical Engineering, State Key Laboratory of Crystal Materials, Shandong University, Ji'nan 250100, People's Republic of China. [2]Shandong Provincial Key Laboratory of Chemical Energy Storage and Novel Cell Technology, and School of Chemistry and Chemical Engineering, Liaocheng University, Liaocheng 252000, People's Republic of China. [3]School of Chemistry and Chemical Engineering, Henan Normal University, Xinxiang 453007, People's Republic of China. [4]Department of Chemistry, College of Science, King Saud University, P. O. Box 2455 Riyadh 11451, Saudi Arabia. ✉e-mail: dsun@sdu.edu.cn

organic building units, which may limit the stability on the process of dynamically structural transformation under drastic physical stimulation[22,23]. In addition, the further post-transformation of such OFs must be done through the redesign of the building unit. Therefore, the development of suitable building unit to rationally construct the C-H···π interaction-stacked porous supramolecular framework (PSF) with both dynamic structure and post-transformation is an important but currently challenging subject.

To address these challenges, we designed a three-dimensional (3D) PSF, namely **Cu12a-π**, based on non-covalent C-H···π interactions between cluster-based building units that are protected by 9-(prop-2-yn-1-yl)−9H-carbazole (Cbz-PrAH) and *p-tert*-butylthiacalix[4]arene (H$_4$TC4A). Of these, the Cbz-PrAH contains extended aromatic ring, which is conducive to the formation of intra- and intermolecular non-covalent C-H···π interactions[24]; H$_4$TC4A is a kind of macrocyclic ligand with intrinsic cavity structure[25-29]. As displayed in Fig. 1, these features of two ligands synergistically lead to construct an as-prepared PSF (**Cu12a**), which can transform to guest-free PSF (**Cu12a-π**) through a dynamical single-crystal-to-single-crystal (SC-SC) fashion by removing guest molecules under heating and vacuum conditions. The monomer Cu$_{12}$ nanocluster as a zero-dimensional (0D) cluster-based building unit, can be further supramolecularly assembled into C-H···π interactions-stacked PSF. Even more impressively, the post-transformation of **Cu12a-π** followed a dissociation-reassembly mechanism, giving the cluster-based nonporous adaptive crystals (NACs) **Cu12b-NACs** with updates of both structure and properties.

## Results

### Synthesis and structural characterization

To implement the synthesis of the as-prepared PSF, **Cu12a**, we selected copper(I) precursor, Cbz-PrACu (Please see experimental details in the Supplementary Methods) to react with H$_4$TC4A via solvothermal method, giving the red rhombic crystals of **Cu12a** in a quite high yield of 91% (Supplementary Fig. 1). The process involved the depolymerization of the starting Cbz-PrACu precursor upon heating. Concomitantly, the H$_4$TC4A was deprotonated, resulting in the formation of 12-nuclei copper(I) nanoclusters protected by Cbz-PrA$^-$ and TC4A$^{4-}$ ligands. The infrared (IR) spectroscopy confirmed the existence of Cbz-PrA$^-$ in **Cu12a**. Besides, elements of **Cu12a** were confirmed by energy-dispersive x-ray (EDX) spectroscopy. More synthetic details

and basic characterizations are listed in the Supplementary Figures and Tables (Supplementary Fig 2–4 and Supplementary Tables 1, 2).

Single-crystal X-ray diffraction (SCXRD) analyses revealed that the monomer of **Cu12a** crystallizes in a monoclinic unit cell under the *P*2$_1$/*c* space group with the asymmetric unit containing a half of nanocluster (Supplementary Fig. 5 and Supplementary Table 1), and its formula is determined as [Cu$_{12}$(Cbz-PrA)$_4$(TC4A)$_2$]·(CH$_3$COOCH$_2$CH$_3$). The structural anatomy of the monomer was shown in Fig. 2. The metal core can be viewed as an approximate cube-like pattern composed of three rectangular Cu$_4$ units rotated by 45° in sequence, the equatorial copper atoms are bulged from the centres of four vertical faces of the cube about 0.446 Å. The Cu$_{12}$ kernel exhibits a $C_4$ symmetry with a $C_4$ axis passing through the centres of two horizontal planes, and is consolidated by short Cu···Cu contacts (average distance of 2.76 Å), which are shorter than the sum of Bondi's van der Waals radii of two Cu atoms (2.80 Å)[30], implying an appreciable cuprophilic interactions. Furthermore, the types of capping ligands in the monomer of **Cu12a** are somewhat reminiscent of the **Cu13a** and **Cu13b**, where Cbz-PrA$^-$ and TC4A$^{4-}$ capped on the periphery of the copper kernel in our previous work[31].

The difference is that four Cbz-PrA$^-$ ligands like nails hold only one coordination mode ($\mu_4$-$\eta^1$: $\eta^1$: $\eta^2$: $\eta^2$, Supplementary Fig. 6a, b), which are evenly distributed on the equator of the kernel, and their carbazole rings are oriented to two directions (up and bottom). The Cu-C bond lengths are in the range of 1.927 ~ 2.241 Å. Apart from the Cbz-PrA$^-$ ligands, the poles of the kernel are clutched by two TC4A$^{4-}$ ligands. The phenolic hydroxyl and bridging sulfur donors of the TC4A$^{4-}$ are bound to four copper atoms, showing the same coordination mode, $\mu_4$-$\kappa_O^2$:$\kappa_O^2$:$\kappa_O^2$:$\kappa_O^2$:$\kappa_S^1$:$\kappa_S^1$:$\kappa_S^1$:$\kappa_S^1$ (Supplementary Fig. 6c). The Cu-O and Cu-S bond lengths related to TC4A$^{4-}$ fall in the ranges of 1.946 ~ 2.390 Å. The four Cbz-PrA$^-$ and two TC4A$^{4-}$ ligands define the basic configuration of the Cu$_{12}$ cluster, and have substantial impacts on the crystal stability and adsorption behavior (*vide infra*).

### Crystalline supramolecular assembly

Recently, the framework adaptability to guest uptake and stimuli-responsiveness are two intriguing features in flexible CPMs that are receiving increasing attentions[32-36]. Inspired by this, the guest EA molecules in **Cu12a** may endow it with similar behavior. Thus, the crystalline sample of **Cu12a** has been heated at 338 K upon vacuum for

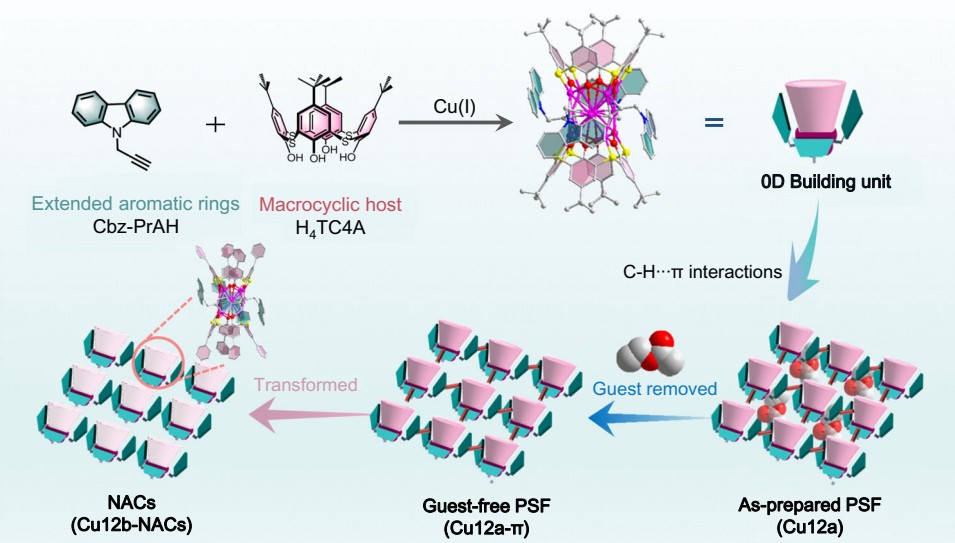

**Fig. 1 | Description of cluster-based PSF.** Cartoon illustration showing the construction of cluster-based building unit. The Cu$_{12}$ cluster is further assembled into PSF via C-H···π interactions; NACs represent the nonporous adaptive crystals; the guest in **Cu12a** is ethyl acetate (EA) molecule.

two days. Surprisingly, the activated sample of **Cu12a**, termed **Cu12a-π**, can still be analyzed via SCXRD, in which the guest EA molecules had been successfully removed to form permanent voids (Fig. 3a and Supplementary Fig. 7). Of note, heating under vacuum, the single crystal of **Cu12a** undergoes a phase transition at 338 K accompanied by a doubling of the crystallographic *a* axis and a change from the *P*2₁/*c* to *C*2/*c* space group. And the two-dimensional (2D) layers of **Cu12a** and

**Cu12a-π** both adopt AA stacking mode, but the latter occurred with a distinct displacement between each layers along the *ab* plane, resulting in a slight reduction in the porosity (from 11.5% to 9.1%). Impressively, upon further soaking **Cu12a-π** in EA solvent for three days, the lattice again recovered to *P*2₁/*c* accompanied with the re-adsorption of the EA molecules. This indicates that the **Cu12a** and **Cu12a-π** as host matrixes can achieve reversible SC-SC transformation by removal and re-adsorption of the guest molecule, thus demonstrating the crystalline structural flexibility of **Cu12a**. Thermogravimetric analysis (TGA) (Fig. 3b) indicates that the EA molecules in **Cu12a** are completely removed after phase transition and the **Cu12a-π** still remains stable even up to 558 K (Fig. 3b). Besides that, the PXRD patterns of **Cu12a** and **Cu12a-π** exhibit distinct difference in peak positions and intensities, further verifying the occurrence of phase transition. Upon heating, the variable-temperature PXRD patterns of **Cu12a-π** show no change even up to 513 K, confirming its thermally robust structure (Fig. 3c), which is superior than most reported πOFs[11–16]. Furthermore, after soaking **Cu12a-π** in different solvents (such as methanol: MeOH, acetonitrile: CH₃CN, ethanol: EtOH, acetone: ACE, n-butanol: *ⁿ*BuOH and n-propanol: *ⁿ*PrOH) for at least 72 h at room temperature, their PXRD patterns were highly consistent well with the theoretical patterns of **Cu12a**, which demonstrated the framework highly dynamic stability to chemical solvents (Fig. 3d). This excellence of thermal and chemical stabilities may be attributed to its unique molecular and packing structural features before and after phase transition.

In Fig. 4a, trumpet-type Cu₁₂ cluster shows significant cavity structures above the poles of copper kernel, which is conductive to the close arrangement of cavity structures in one plane. And the higher rim of the cavity in Cu₁₂ cluster shows a maximum pore diameter of ~8.9 Å. Therefore, Cu₁₂ cluster can be regarded as a porous 0D cluster-based building unit, associated with two kinds of C-H···π interactions (C₍sp3₎-

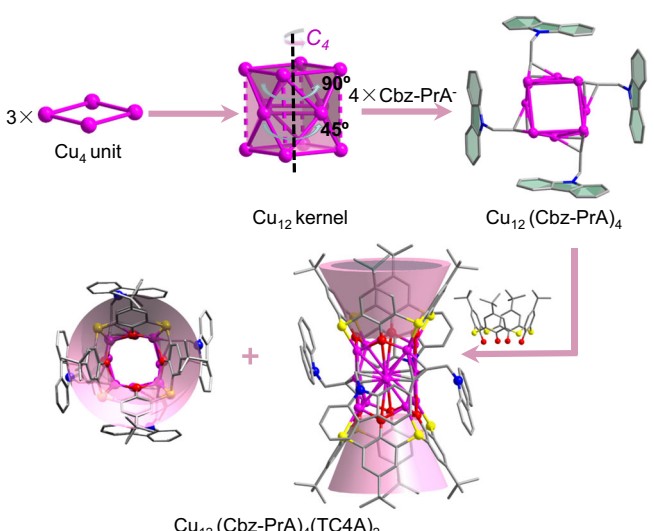

**Fig. 2 | Structural anatomy of the Cu₁₂ cluster.** Hydrogen atoms are removed for clarity. Color labels: purple, Cu; gray, C; red, O; blue, N; yellow, S. The pink cups represent the cavity structure of TC4A⁴⁻.

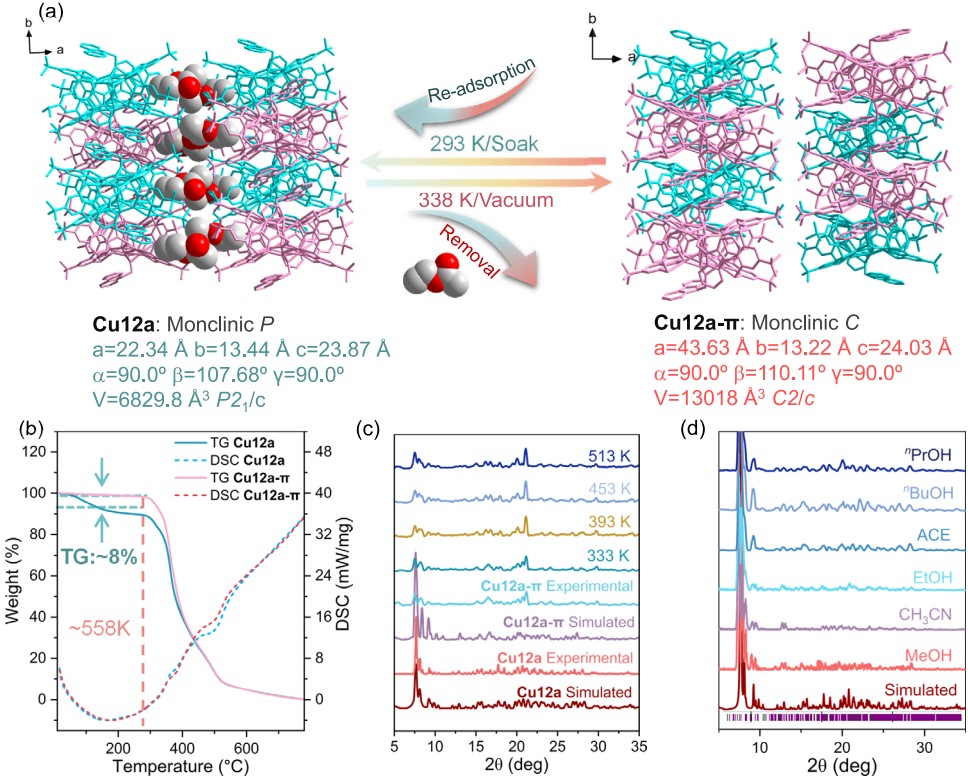

**Fig. 3 | The dynamics of PSF. a** Reversible single-crystal-to-single-crystal transformation between **Cu12a** and **Cu12a-π** and corresponding unit cell parameters. **b** TGA and differential scanning calorimetry (DSC) curves of **Cu12a** and **Cu12a-π**. **c** Variable-temperature PXRD patterns of **Cu12a-π**. **d** PXRD patterns of **Cu12a-π** in different solvents. The guests in **Cu12a** are EA molecules.

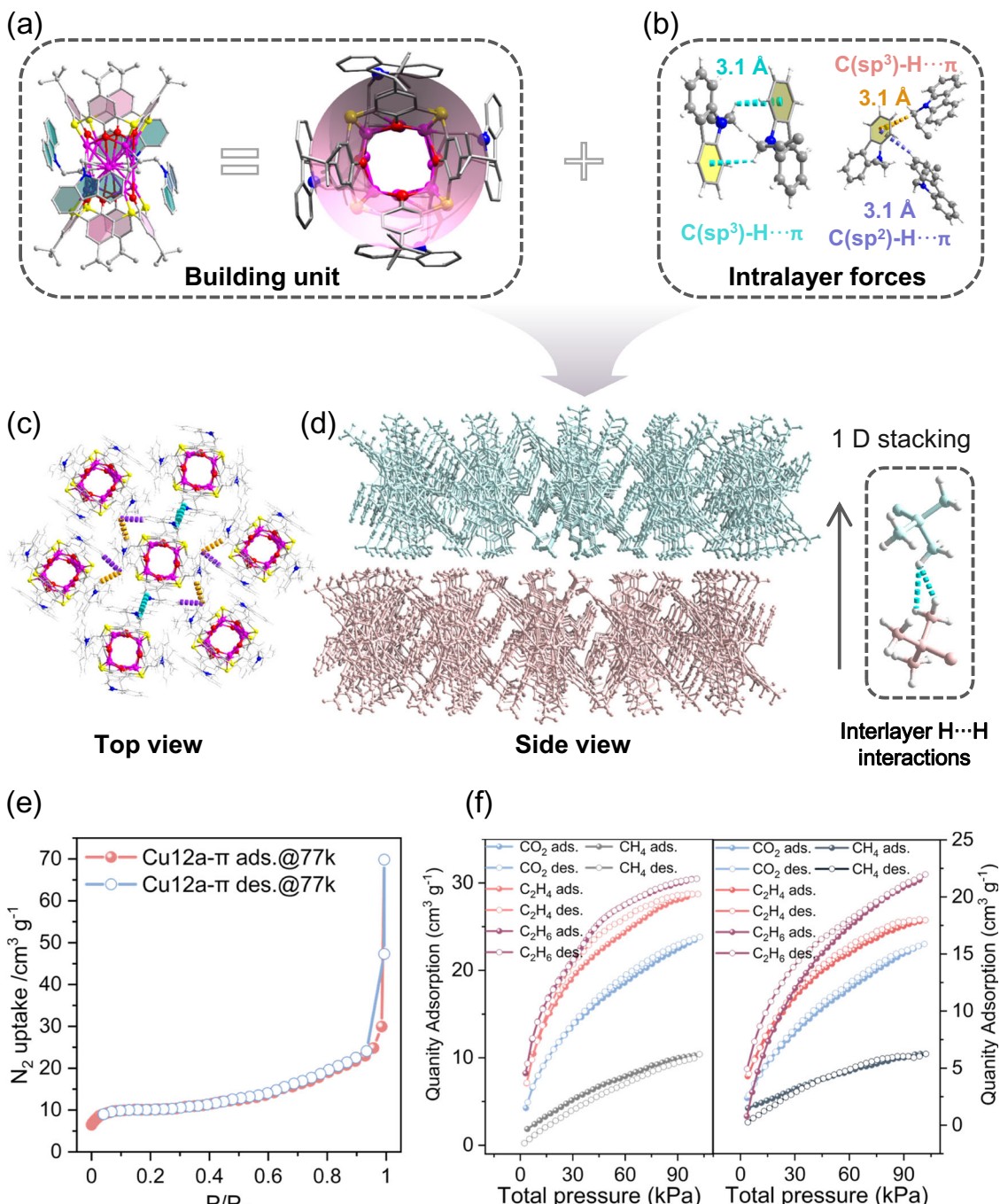

**Fig. 4 | Supramolecular assembly of PSF. a** The $Cu_{12}$ cluster in **Cu12a-π** displays a trumpet-type motif. **b** The intralayer C-H···π interactions between $Cu_{12}$ clusters. The top (**c**) and side (**d**) views of local PSF. **e** Left: $N_2$ adsorption and desorption isotherms of **Cu12a-π** at 77 K; **f** The $CH_4$, $CO_2$, $C_2H_4$ and $C_2H_6$ adsorption-desorption isotherms of **Cu12a-π** at 273 (Left) and 293 K (Right). The sample weight of **Cu12a-π** in gas sorption measurements: 100 mg.

$H$···$π_{(Cbz)}$=3.1 Å and $C_{(sp2)}$-$H$···$π_{(Cbz)}$=3.1 Å) between a carbazole ring and other carbazole ring, as well as between an allylene and a carbazole ring (Fig. 4b), together supramolecular self-assembly into an extended PSF. As shown in Fig. 4c, the $Cu_{12}$ building units adopt 6-connected node to connect six adjacent building units to form a 2D ordered array, which undergoes obliquely slipped one-dimensional (1D) stacking via almost negligible interlayer H···H interactions (2.26 and 2.93 Å) of tertiary butyl to form a final 3D supramolecular framework (Fig. 4d). Of these, the intralayer C-H···π interactions contribute to the framework stability as the dominant linkages compared to the strength of weak interlayer H···H interactions[37]. By contrast, the supramolecular

crystalline assembly of $Cu_{12}$ cluster in **Cu12a** also exhibits a similar mode to **Cu12a-π** (Supplementary Fig. 8), except for the relative large diameter of pore (~9.3 Å) and length of intralayer bridges ($C_{(sp3)}$-$H$···$π_{(Cbz)}$=3.2 or 3.4 Å and $C_{(sp2)}$-$H$···$π_{(Cbz)}$=2.8 Å) due to the incorporation of guest EA solvents.

On the basis of packing characteristic of **Cu12a-π**, it is clearly seen that the small cavities provided by the thiacalix[4]arene and permanent voids between each layers effectively endow the **Cu12a-π** with certain porosity (Supplementary Fig. 7). Correspondingly, analysis of $N_2$ adsorption isotherms of **Cu12a-π** gives rise to a Brunauer-Emmett-Teller (Langmuir) surface area of 31.93 (45.29) $m^2 \cdot g^{-1}$, and a wide pore

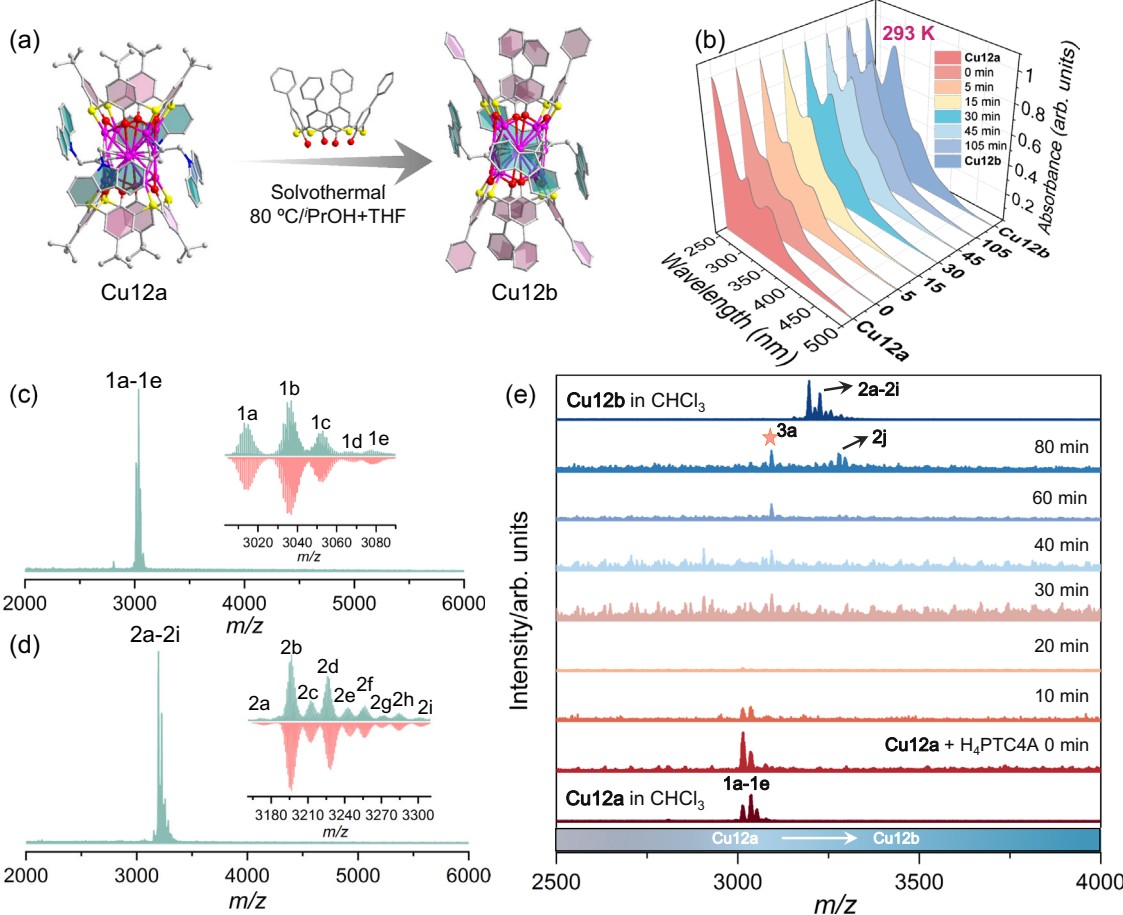

**Fig. 5 | The transformation of PSF. a** Structural transformation of monomer Cu₁₂ cluster from **Cu12a** to **Cu12b**. **b** UV–Vis absorption spectra of the conversion from **Cu12a** to **Cu12b** over the time course at 293 K. **c, d** Positive-ion mode ESI-MS of the crystals of **Cu12a** (**a**) and **Cu12b** (**d**) dissolved in mixed CHCl₃ and CH₃OH. Inset: The zoom-in mass spectrum of experimental (green line) and simulated (red line) isotope-distribution patterns of each labeled species. **e** Time-course ESI-MS of transformation from **Cu12a** to **Cu12b** induced by adding H₄PTC4A.

size distribution in the range of 0.8–1.6 nm calculated by Horvath-Kawazoe (HK) model (Fig. 4e and Supplementary Fig. 9). These results are thought to reflect the small cavities provided by the thiacalix[4] arene and the voids between the 2D layers. Meanwhile, Fig. 4f shows the pure-component equilibrium adsorption isotherms for the $CH_4$, $CO_2$, $C_2H_4$, and $C_2H_6$ collected at different temperatures. The maximum $C_2H_6$ uptake reaches 30.48 $cm^3 \cdot g^{-1}$ (273 K) at normal pressure, further confirming the real existence of permanent porosity in **Cu12a-π**.

## Transformed into NACs

Remarkably, post-modification has been one of the research hotspots for MOFs in recent years, facilitating the acquisition of anticipant structures and properties[38–41]. Inspired by this, the Cu₁₂ nanocluster acting as a building unit may feature ligand-exchange to remodel the PSFs. Therefore, we tried to transform **Cu12a-π** by dissolving crystal-line samples in an isopropanol/tetrahydrofuran (v:v = 3:3) mixture, to which an excess of H₄PTC4A (*p-phenyl*-thiacalix[4]arene) was added, and then the above solution continued to react under solvothermal condition to generate red block crystals of **Cu12b**. Except for the *p-phenyl*-thiacalix[4]arene (Supplementary Fig. 5d), the monomer Cu₁₂ cluster in **Cu12b** has similar compositions to that of **Cu12a**, and the four phenyl groups act as the up rim of cavity structure (Fig. 5a). In view of the significantly different optical absorbance between the two Cu₁₂ clusters (Supplementary Fig. 10a), we carried out the time-course optical absorptions to monitor the conversion process (Fig. 5b). With time going on, the intensity of its characteristic absorption peak at

~300 nm increased significantly upon stirring, as well as the peak at ~340 nm gradually changed from a prominent peak to a shoulder peak. Meanwhile, the color of the mixture solution transformed from yellow to pale yellow and finally to brownish yellow (Supplementary Fig. 10b). So as to intuitively visualize the trend of the two characteristic peaks, the correlation between the absorbance ratio of the two peaks and reaction time was illustrated. As the time went from 0 to 105 min, the value of $A_{300\,nm}/A_{340\,nm}$ was raised from 1.563 to a maximum of 2.361, and then maintained around 2.335 for **Cu12b** (Supplementary Fig. 10c). Considering the following factors: (i) the extra charge transition of $IL_{(Cbz-PrA+PTC4A)}CT$ in **Cu12b** leading to the variation in the absorption intensity of the two nanoclusters at ~300 nm (Please see details in the Supplementary Methods); (ii) the position of the maximum absorption peaks of the two ligands (H₄TC4A and H₄PTC4A) (Supplementary Fig. 10d); (iii) dynamic characteristics of coordination bonds, we tentatively concluded that the increase of absorption strength in $A_{300\,nm}/A_{340\,nm}$ may be attributed to the dissociation of TC4A⁴⁻ and bonding of PTC4A⁴⁻ ligands in **Cu12a**, leading to the increase in the absorption intensity of whole system near 300 nm. When the dissociation and bonding were in equilibrium, the $A_{300\,nm}/A_{340\,nm}$ reached its crest values.

To verify the assumption on the transformation mechanism, we monitored the ligand-exchange process in the structural transformation from **Cu12a** to **Cu12b** by electrospray ionization mass spectrometry (ESI-MS). As shown in Fig. 5c, d, the **Cu12a** and **Cu12b** in mixed CH₃OH and CHCl₃ exhibit a series of +1 species (**1a-1e** for **Cu12a**; **2a-2i** for **Cu12b**) at the *m/z* range of 3010-3090 and 3170-3345, respectively,

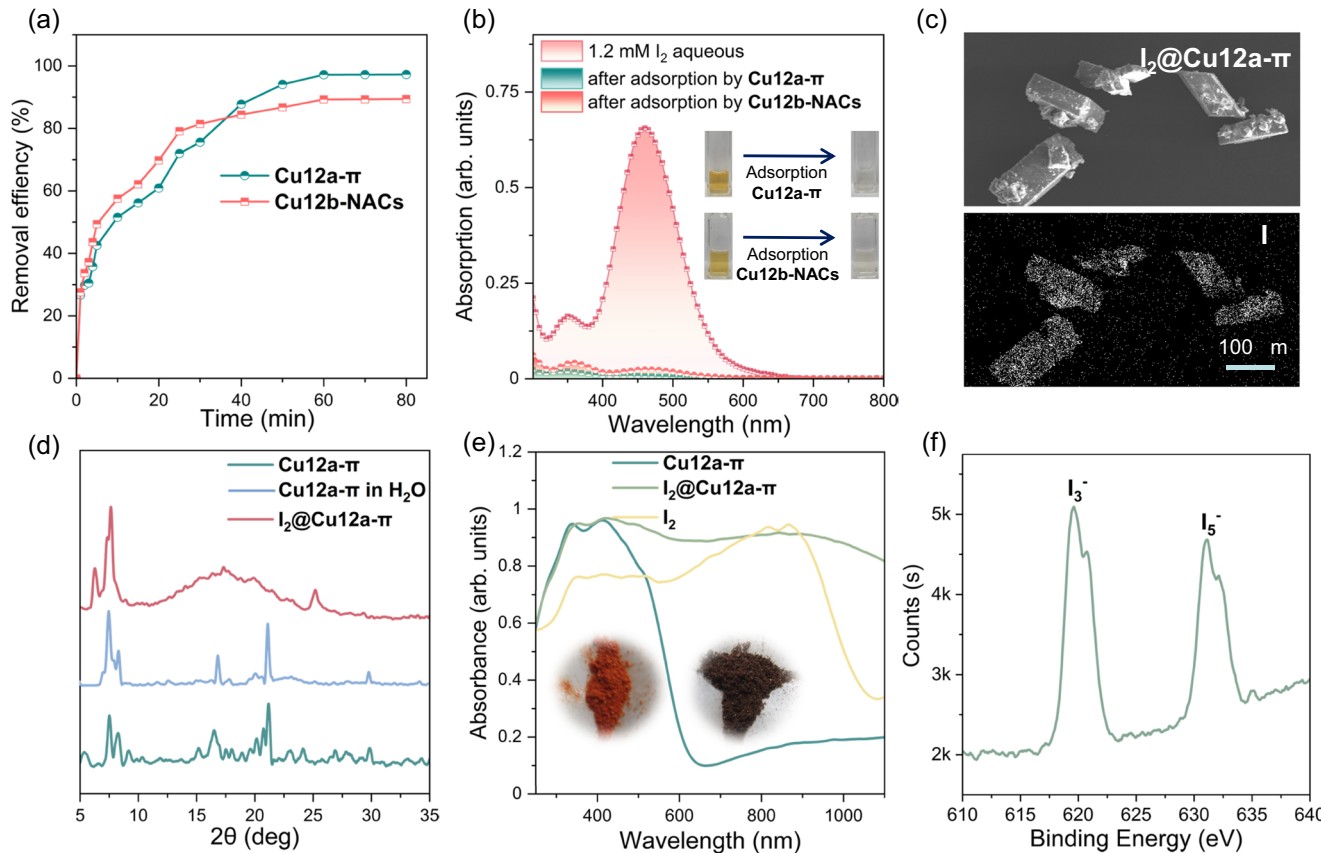

**Fig. 6 | Iodine adsorption of PSF. a** Time-dependent adsorption profiles for the **Cu12a-π** and **Cu12b-NACs** (0.1 mg·mL$^{-1}$) when immersed in 30 mL iodine aqueous solutions (1.2 mM). **b** UV–Vis spectra of saturated iodine aqueous solutions before and after adsorption by **Cu12a-π** and **Cu12b-NACs**, respectively; the inset shows the color changes of the iodine aqueous solution after adsorption by **Cu12a-π** and **Cu12b-NACs**, respectively. **c** SEM and EDX images of **I$_2$@Cu12a-π**, scan bar = 100 μm. **d** PXRD patterns of **Cu12a-π**, **Cu12a-π in H$_2$O** and **I$_2$@Cu12a-π**. **e** UV–Vis spectra of **Cu12a-π**, **I$_2$@Cu12a-π** and **I$_2$**. The inset shows color change of the **Cu12a-π** before (left) and after (right) adsorbing iodine. **f** XPS spectrum of **I$_2$@Cu12a-π**.

which correspond to their intact cluster bound with different cations or solvents. The most dominant peak **1b** at *m/z* 3036.9040 and peak **2b** at *m/z* 3196.6478 can be attributed to [Cu$_{12}$(TC4A)$_2$(Cbz-PrA)$_4$Na]$^+$ (Cal. 3036.8907) and [Cu$_{12}$(PTC4A)$_2$(Cbz-PrA)$_4$Na]$^+$ (Cal. 3196.6410), respectively. However, as the reaction of H$_4$PTC4A and **Cu12a** proceeded in CHCl$_3$, the parent peaks of **Cu12a** (**1a**−**1e**) progressively faded away within 20 min (Fig. 5e), which involves the dissociation of the cluster. Of note, an intermediate species (**3a**) appeared at 30 min, and can be assigned to [Cu$_{12}$(PTC4A)(TC4A)(Cbz-PrA)$_4$H]$^+$. This species **3a** is the mono-substituted product of **Cu12a**, in which one of the TC4A$^{4-}$ ligands is replaced by PTC4A$^{4-}$. After 80 min, we observed a species **2j**, [Cu$_{12}$(PTC4A)$_2$(Cbz-PrA)$_4$(CH$_3$OH)(H$_2$O)$_2$K]$^+$, which can be identified as the bis-substituted product of **Cu12a**. This result indicates that **Cu12b** is more thermodynamically stable than **Cu12a**, which also was confirmed by DFT calculations (Supplementary Fig. 11). Moreover, the collision-induced dissociation (CID) mass further demonstrated the higher stability of **Cu12b** than **Cu12a** in gas phase (Please see details in the Supplementary Figures). More formula assignments for the above species in ESI-MS were performed as listed in Supplementary Table 3.

The transformation between the two Cu$_{12}$ clusters was initially presumed to be a simple ligand-exchange process based on the single crystal structure and optical monitoring. However, further investigation using ESI-MS tracking revealed that this transformation was not a straightforward ligand-substitution, but rather a kinetically driven dissociation of the initial cluster, followed by a reassembly to form a thermodynamically stable new cluster. Thus, we have established the dissociation-reassembly (DR) mechanism to describe above

transformation process, highlighting the dynamic nature of the dissociation and subsequent reassembly steps. The DR mechanism provides a conceptual framework to understand the intricate dynamics involved in the transformation of **Cu12a** to the more stable **Cu12b**[42–50].

Considering the deep cavity of PTC4A$^{4-}$, we further checked the packing formed by Cu$_{12}$ cluster in **Cu12b**. There is one THF molecule residing in the cavity of each PTC4A$^{4-}$ in **Cu12b**, which reduces the void produced by PTC4A$^{4-}$ ligand itself, so the **Cu12b** is a nonporous structure. The activated experiment of **Cu12b** was further carried out, and the activated sample namely **Cu12b-NACs** had been successfully removed the guest THF solvents (Supplementary Fig. 12). Of note, **Cu12b-NACs** also undergoes a SC-SC phase transition accompanied by a trebling of the crystallographic volume. Due to the removal of THF molecules in both lattice and the cavity of PTC4A$^{4-}$, the Cu$_{12}$ clusters are shifted to more closer, producing deeper interpenetration between the upper phenyl rings of adjacent PTC4A$^{4-}$ compared to those in **Cu12b**, where the THF molecules block interpenetration (Supplementary Fig. 13). Thus, the **Cu12b-NACs** can be viewed as cluster-based nonporous adaptive crystals (NACs), which are relatively nonporous but with medium surface areas[51–56]. In light of this, the analysis of N$_2$ adsorption isotherm of **Cu12b-NACs** gives an experimental Brunauer-Emmett-Teller (Langmuir) surface area of 14.88 (12.63) m$^2$·g$^{-1}$, which is less than half of that of **Cu12a-π**, and the pore-size distribution exhibits broad and ambiguous peak (Supplementary Fig. 14), which further verifies that **Cu12b-NACs** acting as the NAC features medium surface area and non-porosity. We also performed the gas adsorption experiments of **Cu12b-NACs** at 273 K, and the uptake amounts of CH$_4$, CO$_2$, C$_2$H$_4$ and C$_2$H$_6$ decreased a half

compared to that of **Cu12a-π**. This observation fits the fact of condensed stacking of crystallographic structure.

## Iodine adsorption

It is well known that capturing iodine with high adsorption capacity from aqueous solution remains a challenge for most porous materials[57–64]. Benefiting from the permanent porosity of **Cu12a-π** and the potentially adaptive absorption capacity of **Cu12b-NACs**, the capacities of iodine uptake would be worthy of further investigation, especially in aqueous solution. Hence, adsorption experiments involved with saturated aqueous solutions of iodine were performed. In this case, the UV–Vis spectra of this aqueous solutions show a weakening of the absorption profile with time in the presence of **Cu12a-π** and **Cu12b-NACs** (Fig. 6a, b and Supplementary Fig. 15–17). The **Cu12a-π** needs much more time to reach equilibrium, but shows higher uptake efficiency compared to **Cu12b-NACs**. Wherein, **Cu12a-π** shows an iodine adsorption efficiency of 97.2% from aqueous solution within 60 min, with a total iodine uptake of 2.96 g·g$^{-1}$, which was superior to most reported iodine adsorbents (Supplementary Table 4). Although, because of the relatively nonporous feature caused by the offset of *p-phenyl-*thiacalix[4]arene on the packing of **Cu12b-NACs**, the uptake efficiency of iodine was about 89.42%, with an equilibrium uptake of 2.73 g·g$^{-1}$. Therefore, we can draw a preliminary conclusion that the iodine adsorption sites of the **Cu12a-π** and **Cu12b-NACs** are not only the cavity of the supramolecular framework, but also are related to the surface areas of crystal. The scanning electron microscopy (SEM) image of **I₂@Cu12a-π** showed the crystal-like morphology (Fig. 6c), and the PXRD pattern of **Cu12a-π** soaked in aqueous solution shows similar diffraction peaks to **Cu12a-π**, indicating its excellent framework stability. Of note, **I₂@Cu12a-π** still shows obvious diffraction peaks, suggesting that the framework of **Cu12a-π** was not interrupted after iodine adsorption (Fig. 6d). Besides, the **Cu12a-π** also shows some degree of recyclable ability and can be further reused for additional iodine adsorption tests via soaking **I₂@Cu12a-π** in EA solvent (Supplementary Fig. 18).

Due to its relatively high iodine adsorption capacity, **Cu12a-π** was chosen as the ideal sample to verify that the iodine was present as polyiodide ions after adsorption. For this, the UV–Vis absorption, energy-dispersive x-ray spectroscopy were carried out. The UV–Vis absorption spectra of **Cu12a-π** and the pristine iodine showcase a significantly different absorption profile compared to **I₂@Cu12a-π** (Fig. 6e) in the solid state. Both the **Cu12a-π** and pristine iodine exhibit weak absorption intensity beyond 900 nm, yet the latter exhibits a broad tail peak over the 1000 nm. The strong absorption of **I₂@Cu12a-π** in near-infrared region may be attributed to the charge transfer between polyiodide anions (mainly $I_3^-$ and $I_5^-$) and **Cu12a-π**, which has been reported previously for iodine adsorption materials[65,66]. Moreover, the $I_{3d}$ X-ray photoelectron spectrum (XPS) of **I₂@Cu12a-π** was measured, and two broad peaks at ~620 and ~632 eV indicate the presence of the $I_3^-$ and $I_5^-$ anions, respectively (Fig. 6f).

## Discussion

In summary, we have successfully developed a dynamic and transformable 3D PSF (**Cu12a-π**) with cluster-based building units held together through intralayer non-covalent C-H··π interactions. The **Cu12a-π** is successfully constructed through the SC-SC transformation by the removal of guest molecules from pristine **Cu12a**, showing unexpected structural dynamics and superior thermal stability than most reported π-stacked frameworks. Moreover, the changing in upper rim of thiacalix[4]arene, **Cu12a-π** can be transformed into the nonporous adaptive crystal **Cu12b-NACs** following a DR mechanism, realizing the structural and performance updates. Based on the permanent porosity, **Cu12a-π** can absorb iodine in aqueous solution reaching a high capacity of 2.96 g·g$^{-1}$. Overall, all unique structural features as well as the iodine adsorption performance of such cluster-

based PSF are indicative of a bright future of this type of porous materials.

## Methods
### Syntheses of the As-prepared PSFs
**Cu12a.** Cbz-PrACu (5.00 mg, 0.02 mmol) and H₄TC4A (13.0 mg, 0.02 mmol) were mixed in 6.00 mL methanol/ethyl acetate (v:v = 3:3). The resulting suspension was sealed and heated at 80 °C for 2000 min. After cooling to room temperature, red crystals of **Cu12a** were formed (4.57 mg; yield: 91%). Selected IR peaks (cm$^{-1}$): 2950 (s), 2906 (w), 2860 (w), 1732 (s), 1584 (m), 1450 (s), 1364 (m), 1315 (m), 1257 (s), 1208 (m), 1037 (m), 926 (w), 885 (w), 840 (s), 751 (s), 711 (s), 625 (w), 531 (m).

**Cu12b.** The crystals of **Cu12a** (4.57 mg, 1.52 μmol) were dissolved in mixed solvents of 6.00 mL isopropanol/tetrahydrofuran (v:v = 3:3), then H₄PTC4A (16.0 mg, 0.02 mmol) was added successively. The resulting suspension was sealed and heated at 80 °C for 2000 min. After cooling to room temperature, red crystals of **Cu12b** were formed (2.07 mg; yield: 43%). Selected IR peaks (cm$^{-1}$): 2955 (s), 2901 (w), 2865 (w), 1777 (w), 1589 (m), 1454 (s), 1364 (m), 1311 (s), 1255 (s), 1202 (m), 916 (w), 876 (w), 831 (s), 741 (s), 714 (s), 616 (w), 526 (m).

## Data availability

The data that support the findings of this study are available within the article and its Supplementary Information files. Other relevant data are available from the corresponding author upon request. The raw data for the TGA, DSC, N₂ adsorption and desorption isotherms are provided with Supplementary Data 1-3. Source data depicting the coordinates of the optimized structures are present Source Data file. The X-ray crystallographic coordinates for structures reported in this article have been deposited at the Cambridge Crystallographic Data Centre, under deposition numbers CCDC: 2234029-2234033 for **Cu12a**, **Cu12a-π**, **Cu12a-readsorption**, **Cu12b** and **Cu12b-NACs**. These data can be obtained free of charge from the Cambridge Crystallographic Data Centre via www.ccdc.cam.ac.uk/data_request/cif. Source data are provided with this paper.

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

## Acknowledgements
This work was financially supported by the National Natural Science Foundation of China (Grant Nos. 22171164, 22325105 and 52261135637 to D.S., 22201159 to Z.W.), the Fok Ying Tong Education Foundation (171009), the Natural Science Foundation of Shandong Province (Nos. ZR2019ZD45, ZR2020ZD35, JQ201803, and ZR2017MB061), the Taishan Scholar Project of Shandong Province of China (Nos. tsqn201812003 and ts20190908), the National Postdoctoral Innovative Talents Support Program (No. BX2021171), China Postdoctoral Science Foundation (No. 2021M700081).

## Author contributions
Original idea was conceived by D.S., experiments and data analyses were performed by C.Z., Z.W., W.-D.S., H.C., L.Z., T.L., X.-Q.H., Z.-Y.G., C.-H.T., M.A., P.C. and D.S., structure characterizations were performed by C.Z. and D.S., manuscript was drafted by C.Z. and D.S. All authors discussed the results and commented on the manuscript.

## Competing interests
The authors declare no competing interests.
