## [Peer Review File · Nature Communications]

Dynamic and Transformable Cu₁₂ Cluster-Based C-H•••π-Stacked Porous Supramolecular FrameworksReviewers' Comments:

Reviewer #1:

Remarks to the Author:

In this manuscript, Sun et. al. report a dynamic CH... π -stacked framework which can be modified by post-treatment. This material shows interesting dynamic behaviors stimulated by solvent removal and incorporation. The modifiable cluster may motivate other researchers to adjust properties of other clusters by post-modification. In addition, this 2D- π OF shows high performance of hydrocarbon separation and iodine capture properties. In my opinion, these merits make it suitable for publication in Nat. Commun. However, the following issues should be addressed properly.

1. The porous 2D- π OF is flexible and the hydrocarbon isotherms seem to be stepwise. Is there any hysteresis between the adsorption and desorption branches?
2. The material shows great iodine uptake performance. Is the material can be recycled? What's the status of iodine in the framework? Can it be located by crystallography?
3. Supplementary Figs 16–21 are not cited in the main text.
4. Some recent references related to selective hydrocarbon adsorptions should be added.
5. The writing of the manuscript should be greatly improved. Please check carefully. There are some errors throughout the manuscript. For example, line 97 "The metal core of can be", line 110 "The Cu-C bond lengths" Cu-O?

Reviewer #2:

Remarks to the Author:

Di Sun and co-workers report the synthesis and characterization of a family of compounds based on the assembly of copper cluster-based molecules. Thus, with the combination of copper with 9-(prop-2-yn-1-yl)-9H-carbazole (Cbz-PrAH) and p-tert-butylthiacalix[4]arene (H4TC4A), they have first crystallized a molecular compound with presence of solvent molecules, which can be subsequently removed by vacuum heating while maintaining the single-crystals intact. Following, the authors also show that the compound can be modified by exchanging the p-tert-butylthiacalix[4]arene ligands by p-tert-phenylthiacalix[4]arene ligands.

Generally speaking, the experimental work, including the synthesis and characterization of the products and the crystallographic studies has been adequately conducted. However, in my opinion, the importance of the reported results is limited, and does not clearly represent a significant advancement or novelty as to justify the publication in Nature Communications. This is based on the following:

The disclosed structures are supramolecular assemblies of cluster-containing molecules. The authors explain how the supramolecular interactions are based on π interaction, but the paper does not provide a rationalization of the significance of the selection of these interaction to design or create novel compounds. In other words, due to their nature the employed organic ligands will inevitably lead to the appearance of π interactions during the crystallization, but it is not clear how this might represent any advantage regarding structural design or properties as compared to other classes of reported materials based on different interactions, either covalent or weak.

The authors have chosen "cluster based 2D CH- π stacked organic frameworks" to label their compounds. However, these cannot be considered as organic compounds, since they are made of both metal atoms and organic molecules.

The use of two-dimensional frameworks also seems arbitrary and not justified, since these are molecular, non-extended compounds, and there is no indication that they could be isolated as 2D materials.

The modification of the compounds is compared with the post-synthetic modification procedures that can be carried out with MOFs (I assumed that by post-modifiable the authors refer to post-synthetic modifications). MOFs can be chemically modified after synthesis while preserving their structural framework integrity ("crystals as molecule"). However, the chemical transformation here reported takes place with molecules in solution rather than with solid crystals. These are rather standard reactions, and not comparable to MOFs post-synthetic chemical modification procedures.

The potential application of the compounds for gas separation is also highly limited, and not fully demonstrated in the manuscript. The sorption capacity of the materials has been tested with single-component equilibrium isotherms. However, differences in gas uptake cannot be directly extrapolated to separation performance of gas mixtures, as these are kinetic processes. In relation to the hydrocarbon gas sorption measurements, pressure values should be given with absolute units rather than relative pressure in the isotherms curve. Also, considering that the compounds are synthesized in small quantity, the authors should also specify the amount used for the collection of the sorption isotherms in the experimental details. The values provided for pore sizes are not consistent with the values expected from the structural analysis. These values are largely overestimated, which in my opinion is due to an artifact arising from the data treatment, and they are not representative from the actual structural features.

In summary, in my opinion the new compounds here reported might be of interest for the supramolecular chemistry and crystal engineering communities, and thus the paper will be more suitable for a more specialized journal rather than for one with broad audience and general interest.

Reviewer #3:

Remarks to the Author:

In this paper, Sun and coworkers reported cluster-based C-H... π -stacked organic frameworks via dense inter-cluster C-H... π stacking interactions. The frameworks act as adaptive crystals on removing and absorbing guest. Furthermore, these two compounds showed the ability to absorb I₂ from aqueous solution. The adsorption capacity for Cu12a-2D π OF and Cu12b-2D π OF are 3.13 g·g⁻¹ and 2.87 g·g⁻¹, respectively. Nevertheless, there are some of the results should be more carefully interpreted and solid experimental section need to be provided. Thus, I recommend its publication in Nature Communications after minor revision.

1. The authors said that "the framework highly dynamic stability to chemical solvents", and on the account of that the iodine absorb was tested in aqueous solution, what about the stability in water for Cu12a-2D π OF?

2. The authors mentioned that "To our knowledge, only the cluster-based π OFs with permanent porosity are true to its name, but they are rare up to now." Please add some literature to support this opinion.

3. This paper points out that it's hard to be elucidated process of conversion and corresponding mechanisms between two Cu12 building units. The energy of those two compounds can be calculated, maybe the lower energy of Cu12b makes the reaction take place. Or, it's the excess of H4PTC4A that makes it more possible to coordination with Cu because of the coordination bond is dynamic.

4. Now that those two compounds can dissolve in dichloromethane, Nuclear Magnetic Resonance Spectroscopy (NMR) can be used to monitor the conversion process which is clearer and exactly than UV-Vis absorption spectrum. On the other hand, the analysis of the UV-Vis absorption spectrum in supporting information can be mentioned in "Post-Modification of 2D π OFs" part. The authors figured out that the excitation of Cu12a at 297.75 nm primarily arise from transition out of the HOMO-1, HOMO-2 and HOMO-5 dominated by TC4A4- ligand into LUMO+6, LUMO+8 and LUMO+11 dominated by Cu12 core and Cbz-PrA- ligand, which comprise the L(TC4A)M(Cu)CT and L(TC4A)L(Cbz-PrA)CT transitions, please give more details about the electrons transition in n , n^* and d orbitals.

5. The description of Cu12b is similar to a class of solid materials for adsorption and separation the nonporous adaptive crystals (NACs), which function at the supramolecular level (J. Am. Chem. Soc. 2022, 144, 113–117; Chem 2023, 9, 1-9), thus, this kind of materials can be mentioned.

6. How to prepare the polyiodide anions in the solid state to test UV-Vis absorption? And why the UV-Vis absorption spectra of Cu12a-2D π OF and the polyiodide anions show a similar absorption peak at 300-500 nm?

7. Except for the cavity of this crystal contributes to the absorption of I₂, are there any other interaction that make Cu12a exhibits high iodine adsorption capacity? Will I₂ react with Cu?

8. There are some details need to be proved. The format of caption needs to be the same. In

supporting information, the figure in "It was apparent that both Cu12a and Cu12b featured similar electronic structures (Supplementary Fig. 16)" should be Supplementary Fig. 17.

Reviewer #1 (Remarks to the Author):

In this manuscript, Sun et. al. report a dynamic CH₂·π-stacked framework which can be modified by post-treatment. This material shows interesting dynamic behaviors stimulated by solvent removal and incorporation. The modifiable cluster may motivate other researchers to adjust properties of other clusters by post-modification. In addition, this 2D-πOF shows high performance of hydrocarbon separation and iodine capture properties. In my opinion, these merits make it suitable for publication in Nat. Commun. However, the following issues should be addressed properly.

Response: We are very pleased and excited by the positive comments on the novelty and significance of our study. We also thank the reviewer's detailed technical comments/suggestions, which help us to improve the manuscript. We have addressed them point-by-point below.

1. The porous 2D-πOF is flexible and the hydrocarbon isotherms seem to be stepwise. Is there any hysteresis between the adsorption and desorption branches?

Response: Thank you for pointing this out! We agree with you that there is hysteresis between the adsorption and desorption branches. The flexible porous materials, also known as soft porous materials, combine the regularity and flexibility and show dynamic frame structure changes driven by the stimulus of guest molecules (*J. Am. Chem. Soc.* 2017, 139, 7733–7736; *Angew. Chem. Int. Ed.* 2022, 61, e202201766; *Natl Sci. Rev.* 2018, 5, 907–919; *Nat. Commun.* 2021, 12, 4097). In this work, Cu12a-π as a flexible porous supramolecular framework (PSF) with flexible framework and cluster-based building unit (Cu12a), should exhibit hysteresis loops between the adsorption and desorption branches. Thus, as you suggested, we have provided the whole process of N₂ adsorption and desorption experiments and the pure-component adsorption isotherms of the four gas molecules. As shown in the Figure 4e, the analysis of N₂ adsorption isotherm of Cu12a-π gives rise to a type IV isotherm with a small hysteresis loop between its adsorption and desorption branches, which is consistent with the mesopores (macrocyclic cavity and voids between layers) and flexibility of framework. Such a hysteresis loop was also observed in the pure-component absorption-desorption cycles of other gas molecules, especially the C₂H₄ and CO₂ (Figure 4f).

New comments have been added in the revised manuscript and Supplementary Materials as: *“Correspondingly, analysis of N₂ adsorption isotherms of Cu12a-π gives rise to a type IV curve with a small hysteresis loop associated with desorption, which indicates the existence of mesopore structure and the breathing effect of framework.”* and *“Moreover, the similar hysteresis loop was also observed in the other gas molecule adsorption and desorption. Fig. 4f shows the pure-component equilibrium adsorption isotherms for the CH₄, CO₂, C₂H₄ and C₂H₆ collected at different temperatures and an expected hysteresis loop in an absorption-desorption cycle, especially for the CO₂ and C₂H₄.”* (Page 7)

Fig. 4 Supramolecular assembly of PSF. **a** The Cu₁₂ cluster in Cu12a-π displaying a trumpet-type motif. **b** The intralayer C-H...π interactions between Cu₁₂ clusters. **c**, **d** The top (**c**) and side (**d**) views of local PSF. **e** Left: N₂ adsorption and desorption isotherms of Cu12a-π at 77 K; **f** Single-component sorption isotherms of the CH₄, CO₂, C₂H₄ and C₂H₆ adsorption and desorption

isotherms of $\text{Cu12a-}\pi$ at 273 (Left) and 293 K (Right). The sample weight of $\text{Cu12a-}\pi$ in gas sorption measurements: 100 mg.

2. The material shows great iodine uptake performance. Is the material can be recycled? What's the status of iodine in the framework? Can it be located by crystallography?

Response: Thanks for your constructive suggestions.

(1) As you suggested, the iodine uptake efficiency of $\text{Cu12a-}\pi$ is excellent, it is significant to investigate the recyclability of such materials. In the revised stage, we carried out the de-sorption experiments of $\text{I}_2@\text{Cu12a-}\pi$ by immersing sample in different solvents, for instance, ethanol (EtOH), acetonitrile (CH_3CN) and ethyl acetate (EA). Among of them, EA can effectively induce the release of iodine from $\text{I}_2@\text{Cu12a-}\pi$ accompanying the color of sample changing from brown to orange. And the dry powder can be used to additional iodine adsorption tests.

New comments and figure have been added in the revised manuscript and **Supplementary Materials** as: "Besides, the $\text{Cu12a-}\pi$ also shows some degree of recyclable ability and can be further reused for additional iodine adsorption tests via soaking $\text{I}_2@\text{Cu12a-}\pi$ in EA solvent." (Page 12)

Supplementary Fig. 18 Color changes of $\text{I}_2@\text{Cu12a-}\pi$ soaked in several solvents (EtOH, CH_3CN and EA) for 30 minutes.

(2) To determine the status of iodine in the framework, we tried to gain a full

understanding of solid-state structure of $I_2@Cu12a-\pi$ by single-crystal X-ray crystallography, but it was hampered by difficulties in obtaining high-quality crystals. Therefore, we used X-Ray photoelectron spectroscopy (XPS), which does not require high crystal quality, to investigate the status of iodine in the framework. As shown in Figure 6f, the X-ray photoelectron spectrum of $I_2@Cu12a-\pi$ showed two broad peaks at ~ 620 and ~ 632 eV, which were attributed to the I_3^- and I_5^- anions, respectively. Therefore, the iodine in the framework are polyiodide anions (mainly I_3^- and I_5^-).

New comments have been added in the revised manuscript as: "Moreover, the I_{3d} X-ray photoelectron spectrum (XPS) of $I_2@Cu12a-\pi$ was measured, and two broad peaks at ~ 620 and ~ 632 eV indicate the presence of the I_3^- and I_5^- anions, respectively (Fig. 6f)." (Page 13)

Fig. 6 **a** Time-dependent adsorption profiles for the $Cu12a-\pi$ and $Cu12b-NACs$ (0.1 mg mL^{-1}) when immersed in 30 mL iodine aqueous solutions (1.2 mM). **b** UV-Vis spectra of saturated iodine aqueous solutions before and after adsorption by $Cu12a-\pi$ and $Cu12b-NACs$, respectively; the inset shows the color changes of the iodine aqueous solution after adsorption by $Cu12a-\pi$ and $Cu12b-NACs$, respectively. **c** SEM image and EDS of $I_2@Cu12a-\pi$, scan bar= $100\ \mu\text{m}$. **d** PXRD patterns of $Cu12a-\pi$, $Cu12a-\pi$ in H_2O and $I_2@Cu12a-\pi$. **e** UV-Vis spectra of $Cu12a-\pi$,

$I_2@Cu12a-\pi$ and I_2 . The inset shows color change of the $Cu12a-\pi$ before (left) and after (right) adsorbing iodine. f XPS spectrum of $I_2@Cu12a-\pi$.

3. Supplementary Figs 16–21 are not cited in the main text.

Response: Thanks for your constructive suggestion. In the initial manuscript, the Supplementary Fig. 16–21 as the additional supporting information have been provided in the Supplementary Information, including the analysis of the electronic structures of the monomers Cu12a and Cu12b through TD-DFT calculation, and the surfaces of Cu12a and Cu12b calculated via 3V volume assessor program. According to your kind reminder, we have cited these Supplementary Figs into the revised manuscript.

New comments have been added in the revised manuscript as: *“More synthetic details and basic characterizations are listed in the Supplementary Information (Supplementary Fig 2-4 and Table 1, 2).” (Page 4) and “Combining the following consideration factors: (i) the extra charge transition of $IL_{(Cbz-PrA+PTC4A)}CT$ in **Cu12b** leading to the variation in the absorption intensity of the two nanoclusters at ~300 nm (see the details in the Supplementary Information; Supplementary Fig. 19-22).” (Page 9)*

4. Some recent references related to selective hydrocarbon adsorptions should be added.

Response: Thanks for your constructive suggestion. In the revision stage, the sorption capacity of the materials has been tested with single-component equilibrium isotherms for CH_4 , C_2H_4 and C_2H_6 . However, differences in gas uptake cannot be directly extrapolated to separation performance of gas mixtures, as these are kinetic processes. Therefore, we have further investigated the breakthrough measurement for mixing gas molecules, yet the separation effect is not obvious. Thus, we decide to delete the inaccurate statement of selective hydrocarbon adsorptions in the revised manuscript, and add this study to the supporting proof section related to pore structure flexibility. Meanwhile, we have added the recent references related to the pore structure flexibility of porous materials in the revised manuscript.

New references have been added in the revised manuscript as:

*39. Elsaidi, S. K., Mohamed, M. H., Banerjee, D. & Thallapally, P. K. Flexibility in Metal–Organic Frameworks: A Fundamental Understanding. *Coord. Chem. Rev.* **358**, 125-152 (2018).*

40. Furtado, F.; Galvoas, P.; Goncalves, M.; Kopinke, F.-D.; Naumov, S.; Rodríguez-Reinoso, F.; Roland, U.; Valiullin, R. & Kärger, J. Guest Diffusion in Interpenetrating Networks of Micro- and Mesopores. J. Am. Chem. Soc. 133, 2437-2443 (2011). (Page 17)

5. The writing of the manuscript should be greatly improved. Please check carefully. There are some errors throughout the manuscript. For example, line 97 "The metal core of can be", line 110 "The Cu-C bond lengths" Cu-O?

Response: We sincerely thank the reviewer for pointing this out. We have carefully checked the full manuscript and corrected the flaws and mistakes. We believe that the quality of revised manuscript has been improved.

The sentence "The metal core of can be viewed as an approximate cube-like pattern composed of three rectangular Cu₄ units rotated by 45° in sequence." have been corrected to "The metal core can be viewed as an approximate cube-like pattern composed of three rectangular Cu₄ units rotated by 45° in sequence."(Page 4)

We will appreciate it if the reviewers can understand our endeavor! Thank you very much for your insightful suggestion!

Reviewer #2 (Remarks to the Author):

Di Sun and co-workers report the synthesis and characterization of a family of compounds based on the assembly of copper cluster-based molecules. Thus, with the combination of copper with 9-(prop-2-yn-1-yl)-9H-carbazole (Cbz-PrAH) and *p*-tert-butylthiacalix[4]arene (H₄TC4A), they have first crystallized a molecular compound with presence of solvent molecules, which can be subsequently removed by vacuum heating while maintaining the single-crystals intact. Following, the authors also show that the compound can be modified by exchanging the *p*-tert-butylthiacalix[4]arene ligands by *p*-tert-phenylthiacalix[4]arene ligands.

Generally speaking, the experimental work, including the synthesis and characterization of the products and the crystallographic studies has been adequately conducted. However, in my opinion, the importance of the reported results is limited, and does not clearly represent a significant advancement or novelty as to justify the publication in Nature Communications. This is based on the following:

Response: We are very pleased and excited by such positive comments on the novelty and significance of this work. We are also very grateful for the reviewer's meticulous and valuable comments to help us improve this manuscript to a much better scientific level. Some important scientific advances in this work are re-summarized as below:

- (1) By reviewing the literatures, we noticed that π -stacked organic frameworks (π OFs) with π - π stacking interactions are classic supramolecular compounds compared to the classic crystal porous materials (CPMs), such as metal-organic frameworks (MOFs) and covalent organic frameworks (COFs). Nevertheless, the C-H \cdots π interactions have always been mixed with π - π stacking interactions, giving assistance to support the π OFs (*PNAS* 2020, 117, 20397; *Angew. Chem.* 2015, 127, 579–583; *Angew. Chem.* 2004, 116, 2852–2857). Therefore, it will be meaningful but challenging to investigate the performance of such force in the construction of framework. In addition, organic molecules are considered to be one of the most important building blocks in the construction of these framework materials, but there is inevitably an intrinsic poor stability problem in framework. In contrast, metal nanoclusters used as cluster-based building units will effectively avoid this**

drawback (*Sci. Adv.* 2020, 6, eaax9976). In this work, we first reported a new C-H \cdots π interactions-stacked porous supramolecular framework (PSF), Cu₁₂a- π , which possesses both C-H \cdots π interactions and metal cluster-based building units.

- (2) Second, Cu₁₂a- π shows unique and rare dynamic feature. To our knowledge, the dynamic or flexible MOFs possess alterable degrees of frame freedom, which leads to the breathing or swelling behavior. These frame freedoms are mainly caused by: i) the rotation and conformational change of the organic linker; ii) the organic linker including the swinging or shifting of donor groups (*Angew. Chem. Int. Ed.* 2022, 61, e202201766; *Dalton Trans.* 2016, 45, 4073-4089), which also limits the further expansion and design of dynamic MOF. However, compared to the organic linker in dynamic MOFs, the non-covalent C-H \cdots π interactions are far more labile, but precisely because of their relatively labile inherent properties, they also provide the building units with some freedom. Thus, the framework of Cu₁₂a- π showed dynamic feature via single-crystal-to-single-crystal (SCSC) transformation, exhibiting a certain adaptability to physicochemical stimulation. Meanwhile, the dynamic feature of coordination band allows the PSF with dynamic ligand exchange ability, and thus realizes the transformation of PSF.
- (3) Third, Cu₁₂a- π as cluster-based PSF with flexible and permanent voids shows high performance of iodine capture properties in aqueous solutions, which promotes the application of such PSFs in the treatment of pollutants, especially the radioactive iodine.

In addition, in the revised stage, to address your concerns, we performed more experiments, including breakthrough experiments, and also some new comments and Figures were added according to your suggestions.

We believe that the revised manuscript (with better conveyed scientific contents) improved quality thanks to your constructive suggestions, and would be appealed to heterogeneous readership of *Nature Communications*. We also expect that this work will attract great interest of researchers from inorganic chemistry, cluster chemistry, materials science, nanochemistry and so on.

1. The disclosed structures are supramolecular assemblies of cluster-containing molecules. The authors explain how the supramolecular interactions are based on pi interaction, but the paper does not provide a rationalization of the significance of the selection of these interaction to design or create novel compounds. In other words, due to their nature the employed organic ligands will inevitably lead to the appearance of pi interactions during the crystallization, but it is not clear how this might represent any advantage regarding structural design or properties as compared to other classes of reported materials based on different interactions, either covalent or weak.

Response: We appreciate this question from the reviewer. We agree with the reviewer that the significance of the selection of these interactions to design or create novel compounds should be provided to guide the widespread readers.

- (1) By investigating literatures, we noticed: i) Most of the reported π OFs are not stabilized by pure $\pi \cdots \pi$ stacking interactions, whereas other forces are also involved to a greater or lesser extent, e.g. C-H $\cdots \pi$ stacking. Thus, constructing novel supramolecular networks by C-H $\cdots \pi$ stacking interactions are meaningful to investigate the performance of such a non-covalent interaction in π -stacked PSF. ii) The non-covalent C-H $\cdots \pi$ interactions are relatively weak in strength compared to the coordination bonds of the MOFs or the covalent bonds of COFs, but it is precisely this property that gives the framework stacked by such interactions a certain flexibility. iii) The cluster-based building units have an inherent advantage over organic building units in terms of stability due to their relatively stronger metal-organic bond strength. The stability of cluster-based building units can be improved by the selection of ligands with strong coordination ability, which can effectively compensate for the stability disadvantages of non-covalent interactions-stacked frameworks compared to HOFs/ π OFs materials. Based on the above-mentioned, we decided to design and prepare a robust and dynamic cluster-based C-H $\cdots \pi$ interactions-stacked PSF, which is combined with relatively weak C-H $\cdots \pi$ stacking interactions and stable cluster-building units.
- (2) Towards this final goal (both C-H $\cdots \pi$ stacking interactions and cluster-based building unit with large cavity), we make a great effort in selecting ligands and

preparing C-H \cdots π interactions-stacked PSF. We selected 9-(prop-2-yn-1-yl)-9H-carbazole (Cbz-PrAH) as the main protective ligand because the aromatic rings of Cbz-PrAH on nanoclusters surface would provide an ideal platform to establish intra- and intermolecular interactions, which play a crucial role in stabilizing nanoclusters or their packing in crystals (*J. Mater. Chem. C* 2020, 8, 715–720). Meanwhile, we selected *p*-tert-butylthiacalix[4]arene (H₄TC4A) known as an excellent macrocyclic host to co-protect clusters (*Sci. Adv.* 2016, 2, e1600323), whose intrinsic cavity structure can absorb small molecules via host-guest effect. Based on the above, we obtained a cluster-based C-H \cdots π interactions-stacked PSF with both robust and dynamic advantages. Moreover, precisely because of the two ligands each represent clear functions in their structural design, which gives a very promising strategy for designing this kind of PSF, namely the functionally oriented ligand design strategy. The subsequent redesigns can be realized based on these two functional ligands, for example, the alkyne ligands with more extended aromatic ring and the macrocyclic ligands with larger cavity.

In summary, this work not only validates the significance and performance of such non-covalent interaction in the construction of PSF, but also provides an ideal platform to showcase such functionally oriented ligand design strategy. We believe that the significance of this work will attract great interest of researchers from cluster chemistry, porous materials science.

2. The authors have chosen “cluster based 2D CH- π stacked organic frameworks” to label their compounds. However, these cannot be considered as organic compounds, since they are made of both metal atoms and organic molecules.

Response: We are very grateful to the suggestion from the reviewer. We highly endorse the statements from the reviewer, this C-H \cdots π interactions stacked framework cannot be considered as nOF due to the mixing of copper atoms and organic ligands. Thus, we chose the “C-H \cdots π stacked porous supramolecular framework”, which can be more suitable to label this compound in this work (abbreviate as Cu_{12a}- π) (*Nat. Commun.* 2016, 7, 11564; *Sci. Adv.* 2020, 6, eaax9976). The labels have been corrected in the revised manuscript and Supplementary Information.

3. The use of two-dimensional frameworks also seems arbitrary and not justified, since these are molecular, non-extended compounds, and there is no indication that they could be isolated as 2D materials.

Response: Thanks for the constructive suggestion from the reviewer. As the reviewer's statement, the use of two-dimensional framework are unsuitable for Cu₁₂a- π . Through the in-depth study on two-dimensional porous materials (*Small Methods* 2017, 1, 1600030; *Chem. Soc. Rev.*, 2020, 49, 3920–3951; *J. Am. Chem. Soc.* 2020, 142, 1367–1374; *Chem. Soc. Rev.*, 2018, 47, 6267–6295), we have realized that the cluster-based units hold together via non-covalent interactions along certain layer, but not a non-extended compounds. Therefore, in the revised manuscript, we have corrected the statement of two-dimension, and further investigated the three-dimensional framework of Cu₁₂a- π . In the packing of Cu₁₂a- π , each cluster-based monomer Cu₁₂a is connected to six adjacent Cu₁₂a monomers through two kinds of C-H $\cdots\pi$ interactions through the aromatic rings of Cbz-PrA- ligands. Such packing forms a two-dimensional layer which undergoes one-dimensional stacking through extremely liable short-range interactions (H \cdots H) to form a three-dimensional porous supramolecular framework.

New comments have been added in the revised manuscript as: "Therefore, Cu₁₂ cluster can be regarded as a porous 0D cluster-based building unit, associated with two kinds of C-H $\cdots\pi$ interactions ($C_{(sp^3)}-H\cdots\pi_{(Cbz)}=3.1\text{ \AA}$ and $C_{(sp^2)}-H\cdots\pi_{(Cbz)}=3.1\text{ \AA}$) between a carbazole ring and other carbazole ring, as well as between an allylene and a carbazole ring (Fig. 4b), together supramolecular self-assembly into an extended PSF. As shown in Fig. 4c, the Cu₁₂ building units adopt 6-connected node to connect six adjacent building units to form a 2D ordered array, which undergoes obliquely slipped one-dimensional (1D) stacking via almost negligible interlayer H \cdots H interactions (2.26 and 2.93 Å) of tertiary butyl to form a final 3D supramolecular framework (Fig. 4d)." **(Page 7)**

Fig. 4 Supramolecular assembly of PSF. **a** The Cu_{12} cluster in $\text{Cu}_{12}\text{a}-\pi$ displaying a trumpet-type motif. **b** The intralayer $\text{C}-\text{H}\cdots\pi$ interactions between Cu_{12} clusters. **c**, **d** The top (**c**) and side (**d**) views of local PSF. **e** Left: N_2 adsorption and desorption isotherms of $\text{Cu}_{12}\text{a}-\pi$ at 77 K; **f** Single-component sorption isotherms of the CH_4 , CO_2 , C_2H_4 and C_2H_6 adsorption and desorption isotherms of $\text{Cu}_{12}\text{a}-\pi$ at 273 (Left) and 293 K (Right). The sample weight of $\text{Cu}_{12}\text{a}-\pi$ in gas sorption measurements: 100 mg.

4. The modification of the compounds is compared with the post-synthetic modification procedures that can be carried out with MOFs (I assumed that by post-modifiable the authors refer to post-synthetic modifications). MOFs can be chemically modified after synthesis while preserving their structural framework integrity (“crystals as molecule”). However, the chemical transformation here reported takes place with molecules in

solution rather than with solid crystals. These are rather standard reactions, and not comparable to MOFs post-synthetic chemical modification procedures.

Response: Thanks for the constructive suggestion from the reviewer. We fully agree with the reviewer's suggestion that the transformation of Cu12a- π induced by ligand-exchange cannot be defined as post-synthetic modifications because this transformation is essentially a standard chemical reaction. However, the characteristics of ligand-exchange-induced transformation of clusters can be also be applied to the relatively rapid reconstruction of new cluster-based porous materials as compared to the redesign and synthesis of porous materials. Therefore, as you suggested, we have corrected the "post-modifiable" to "transformable" in the revised manuscript, which will be more appropriate for defining such transformation process.

5. The potential application of the compounds for gas separation is also highly limited, and not fully demonstrated in the manuscript. The sorption capacity of the materials has been tested with single-component equilibrium isotherms. However, differences in gas uptake cannot be directly extrapolated to separation performance of gas mixtures, as these are kinetic processes. In relation to the hydrocarbon gas sorption measurements, pressure values should be given with absolute units rather than relative pressure in the isotherms curve. Also, considering that the compounds are synthesized in small quantity, the authors should also specify the amount used for the collection of the sorption isotherms in the experimental details. The values provided for pore sizes are not consistent with the values expected from the structural analysis. These values are largely overestimated, which in my opinion is due to an artifact arising from the data treatment, and they are not representative from the actual structural features.

Response: Thanks for the constructive suggestion from the reviewer.

(1) As the reviewer's suggestion, the differences in gas uptake cannot be directly extrapolated to separation performance of gas mixtures, thus, we have performed the single dynamic breakthrough curves of equimolar C₂H₆/CH₄ and C₂H₄/CH₄ gas mixtures through a separation column filling with Cu12a- π sample at 298 K and 1 bar. However, the results of experimental breakthrough did not meet our expectations (Figure R1). Thus, we revised the statements related to gas separation

in revised manuscript and used the hysteresis loops that occur in the desorption of gas molecules as supporting evidence for the flexible character of the pore structure of Cu12a- π (Figure 4f).

Fig. R1 Single dynamic breakthrough curves of equimolar CH₄/C₂H₄ (a) and CH₄/C₂H₆ (b) gas mixtures through a separation column filling with Cu12a- π sample at 298 K and 1 bar.

New comments have been added in the revised manuscript as: *“Moreover, the similar hysteresis loop was also observed in the other gas molecule adsorption and desorption. Fig. 4f shows the pure-component equilibrium adsorption isotherms for the CH₄, CO₂, C₂H₄ and C₂H₆ collected at different temperatures, and an expected hysteresis loop in an adsorption-desorption cycle, especially for the CO₂ and C₂H₄. All these results point to the deduction that the mesopores exist in Cu12a- π , whose pore structure can be adapted by changing the guest molecules (solvent or gas molecules). (Page 7)*

- (2) In the gas sorption measurements, the units of pressure values in the isotherms curve have been corrected as absolute pressure instead of relative pressure. Meanwhile, the loading of the Cu12a- π have been added in the revised manuscript to specify the amount used for the collection of the sorption isotherms.

New comments have been added in the caption of revised Fig. 4 as: *“Supramolecular assembly of PSF. (e) Left: N₂ adsorption and desorption isotherms of Cu12a- π at 77 K; (f) Single-component sorption isotherms of the CH₄, CO₂, C₂H₄ and C₂H₆ adsorption and desorption isotherms of Cu12a- π at 273 (Left) and 293 K (Right). The sample weight of Cu12a- π in gas sorption measurements: 100 mg. (Page 8)*

- (3) As for the inconformity between experimental values of pore sizes and the values

expected from the structural analysis, we have repeated the BET experiment three times to resolve the concern of reviewer, yet the results are still quite similar, indicating the reproducibility and reliability of the data. Similar results have been observed in the macrocyclic-based porous materials (*Angew. Chem. Int. Ed.*, 2021, 60, 7188-7196; *Angew. Chem. Int. Ed.*, 2022, 61, e202113724), which is attributed to the existence of flexible mesopore structure and breathing effect of framework.

New comments have been added in the revised manuscript as: "Further experimental Brunauer-Emmett-Teller (Langmuir) surface area of 31.93 (45.29) m²·g⁻¹ was calculated (Supplementary Fig. 9a, b), and the pore size distribution was analyzed based on Horvath-Kawazoe (HK) model, showing wide pore distribution in the range of 0.8–1.6 nm (Supplementary Fig. 9c). Since the *p*-tert-butylthiacalix[4]arene is able to expand its pore-size by adsorbing gases or small molecules, as observed in **Cu13b** in our previous work (Supplementary Fig. 10),³² the wide range of pore distribution mentioned-above may reflect the typically structural flexibility of permanent pores of **Cu12a-π**. This implies that under slightly higher pressures, the permanent pores of **Cu12a-π** can be gradually enlarged, leading to an increase in calculated pore diameter." (Page 7)

We will appreciate it if the reviewers can understand our endeavor! Thank you very much for your insightful suggestion!

Reviewer #3 (Remarks to the Author):

In this paper, Sun and coworkers reported cluster-based C-H \cdots π -stacked organic frameworks via dense inter-cluster C-H \cdots π -stacking interactions. The frameworks act as adaptive crystals on removing and absorbing guest. Furthermore, these two compounds showed the ability to absorb I₂ from aqueous solution. The adsorption capacity for Cu12a-2D π OF and Cu12b-2D π OF are 3.13 g g⁻¹ and 2.87 g g⁻¹, respectively. Nevertheless, there are some of the results should be more carefully interpreted and solid experimental section need to be provided. Thus, I recommend its publication in Nature Communications after minor revision.

Response: We are very pleased and excited by such positive comments on the novelty and significance of this work. We also thank the reviewer's detailed technical comments/suggestions, which we have addressed point-by-point below.

1. The authors said that "the framework highly dynamic stability to chemical solvents", and on the account of that the iodine absorb was tested in aqueous solution, what about the stability in water for Cu12a-2D π OF?

Response: Thanks for your constructive suggestion. The stability of Cu12a- π in aqueous solution is critical for its further performance in iodine adsorption. Therefore, following your suggestion, we have performed the powder X-ray diffraction of Cu12a- π after soaking in water for about 12 h. The result, as expected, shows similar diffraction peaks to Cu12a- π , indicating that Cu12a- π has excellent stability in aqueous solution (Fig. 6d).

New comments have been added in the revised manuscript as: *"and the PXRD pattern of Cu12a- π soaked in aqueous solution shows similar diffraction peaks to Cu12a- π , indicating its excellent framework stability. In addition, I₂@Cu12a- π still shows obvious diffraction peak, suggesting that the framework of Cu12a- π was not interrupted after iodine adsorption (Fig. 6d)."*

(Page 12)

Fig. 6 **a** Time-dependent adsorption profiles for the *Cu12a-π* and *Cu12b-NACs* (0.1 mg mL^{-1}) when immersed in 30 mL iodine aqueous solutions (1.2 mM). **b** UV-Vis spectra of saturated iodine aqueous solutions before and after adsorption by *Cu12a-π* and *Cu12b-NACs*, respectively; the inset shows the color changes of the iodine aqueous solution after adsorption by *Cu12a-π* and *Cu12b-NACs*, respectively. **c** SEM image and EDS of $\text{I}_2@Cu12a-\pi$, scan bar=100 μm . **d** PXRD patterns of *Cu12a-π*, *Cu12a-π* in H_2O and $\text{I}_2@Cu12a-\pi$. **e** UV-Vis spectra of *Cu12a-π*, $\text{I}_2@Cu12a-\pi$ and I_2 . The inset shows color change of the *Cu12a-π* before and after adsorbing iodine. **f** XPS spectrum of $\text{I}_2@Cu12a-\pi$.

2. The authors mentioned that “To our knowledge, only the cluster-based π OFs with permanent porosity are true to its name, but they are rare up to now.” Please add some literature to support this opinion.

Response: Thanks for your constructive suggestion of the reviewer. As many crystalline porous materials are porous in the crystalline state, the crystal may undergo a phase change resulting in the change of porosity upon physical stimulation (*Chem. Mater.* 2015, 27, 6, 1905–1916; *Materials* 2014, 7, 3198–3250; *Angew. Chem. Int. Ed.* 2010, 49, 7526–7529; *Angew. Chem. Int. Ed.* 2004, 43, 5033–5036.). Therefore, we deem that only porous materials with permanent porosity under various physical stimulation are true its name.

In other words, only the π -stacked porous supramolecular framework with permanent porosity are true to its name. However, inspired by the study of nonporous adaptive crystal (NACs) (*J. Am. Chem. Soc.* 2022, 144, 113–117; *J. Am. Chem. Soc.* 2021, 143, 18849–18853), which are relatively nonporous but whose structure can formally adapted to capture guests in the crystalline state, we have realized that the sentence is inaccurate. Thus, we have deleted the less rigorous statements “To our knowledge, only the cluster-based π OFs with permanent porosity are true to its name, but they are rare up to now.” in the revised manuscript. (Page 9)

3. This paper points out that it's hard to be elucidated process of conversion and corresponding mechanisms between two Cu_{12} building units. The energy of those two compounds can be calculated, maybe the lower energy of **Cu12b** makes the reaction take place. Or, it's the excess of $\text{H}_4\text{PTC4A}$ that makes it more possible to coordination with Cu because of the coordination bond is dynamic.

Response: Thanks for your constructive suggestion of the reviewer. In this work, the transformation of the cluster-based building units Cu_{12}a into Cu_{12}b by ligand-exchange gives the opportunity to remodel the framework of $\text{Cu}_{12}\text{a}-\pi$, while its precise mechanism was hard to elucidate. As you suggested, the energy of those two Cu_{12} clusters can be calculated via DFT calculation, which has been reported by several literatures. For example, Wang et. al. calculated the single point energies of isomeric $\text{Au}_{23}\text{-1}/\text{Au}_{23}\text{-2}$ to confirm that the $\text{Au}_{23}\text{-1}$ is energetically favorable over $\text{Au}_{23}\text{-2}$ (*J. Am. Chem. Soc.* 2020, 142, 6, 2995–3001). Of note, the precondition of this calculation is that the two compounds must maintain conservation of chemical formula, otherwise the result will lose the significance of comparison. This is why such calculations are often found in the calculation of isomers and the pathways of catalytic reactions (*J. Am. Chem. Soc.* 2020, 142, 12140–12145; *ACS Catal.* 2022, 12, 15323–15333).

In this work, we have also performed the time-dependent optical absorptions of conversion process under room/low temperature and TD-DFT calculation to confirm the possible mechanism of thermodynamic stabilization driving (or stabilizer exchange) (*Chem. Mater.* 2019, 31, 9939–9969). Besides, we believe the dynamic characteristics of coordination bonds between copper and O/S atoms may also be an important factor in

facilitating the conversion.

New comments have been added in the revised manuscript as: *“Combining the following consideration factors: (i) the extra charge transition of $IL_{(Cbz-PrA+PTC4A)}CT$ in **Cu12b** leading to the variation in the absorption intensity of the two nanoclusters at ~ 300 nm (see the details in the Supplementary Information; Supplementary Fig. 19-22); (ii) the position of the maximum absorption peaks of the two ligands (H_4TC4A and H_4PTC4A) (Supplementary Fig. 11b); (iii) dynamic characteristics of coordination bonds, we tentatively concluded that the increase of absorption strength in $A_{300\text{ nm}}/A_{340\text{ nm}}$ may be attributed to the dissociation of $TC4A^{4-}$ and bonding of $PTC4A^{4-}$ ligands in **Cu12a**, leading to the increase in the absorption intensity of whole system near 300 nm.” (Page 10)*

4. Now that those two compounds can dissolve in dichloromethane, Nuclear Magnetic Resonance Spectroscopy (NMR) can be used to monitor the conversion process which is clearer and exactly than UV-Vis absorption spectrum. On the other hand, the analysis of the UV-Vis absorption spectrum in supporting information can be mentioned in “Post-Modification of 2D π OFs” part. The authors figured out that the excitation of Cu12a at 297.75 nm primarily arise from transition out of the HOMO-1, HOMO-2 and HOMO-5 dominated by $TC4A^{4-}$ ligand into LUMO+6, LUMO+8 and LUMO+11 dominated by Cu12 core and Cbz-PrA- ligand, which comprise the $L(TC4A)M(Cu)CT$ and $L(TC4A)L(Cbz-PrA)CT$ transitions, please give more details- about the electrons transition in π , π^* and d orbitals.

Response: Thanks for your constructive suggestion of the reviewer.

(1) Firstly, we agree with you that Nuclear Magnetic Resonance Spectroscopy (NMR) can be utilized to monitor the conversion process. In our efforts to track the conversion from Cu12a to Cu12b using NMR, we did not obtain the useful information from the NMR spectrum due to the lower solubility of the two clusters in dichloromethane. Nevertheless, electron absorption spectroscopy was also known as a direct and effective tool to monitor the conversion process of nanocluster, for instance, the solvent/ligand-induced structural transformation (*J. Am. Chem. Soc.* 2020, 142, 6, 2995–3001; *J. Am. Chem. Soc.* 2020, 142, 28, 12140–12145; *Chem. Sci.*, 2019,

10, 8685; *Chem. Mater.* 2021, 33, 1, 39–62.). Since the notable variations in absorption spectra between the two Cu₁₂ clusters, particularly in the absorption intensities at approximately 300 nm, it is conducive to track the conversion process between them. Fortunately, the time-dependent optical absorption provides an effective means of illustrating the alterations in electronic structure that occur during the transformation from Cu12a to Cu12b, and sheds light on the potential mechanisms underlying this transformation.

- (2) Secondly, we strongly endorse your proposal to refer the analysis of the UV-Vis absorption spectrum in supporting information to “post-transformation” part. The absorption spectra of the nanoclusters serve as a visual representation of their electronic structures; hence, their corresponding theoretical calculations are valuable to explain the changes in the absorption peaks during the conversion process.

New comments have been added in the revised manuscript as: *“(i) the extra charge transition of $L_{(Cbz-PrA+PTC4A)}CT$ in Cu12b leading to the variation in the absorption intensity of the two nanoclusters at ~300 nm (see the details in the Supplementary Information; Supplementary Fig. 19-22);” (Page 10)*

- (3) Lastly, as you suggested, we have added the analysis of orbital component analysis related to the corresponding transitions in detail, which will facilitate the understanding of the attribution of charge transitions.

New comments have been added in the revised Supplementary Information as: *“These orbitals are all comprised of the p character from the aromatic rings of TC4A⁴⁻ ligand, atomic d character from Cu₁₂ core, as well as small amount of p character from Cbz-PrA⁻ ligand, which comprise the $L_{(TC4A)}M_{(Cu)}CT$ and $L_{(TC4A)}L_{(Cbz-PrA)}CT$ transitions.” (Page 4)*
“These orbitals of Cu12b show nearly same character with Cu12a, except the extra transition partially locate in π/π^ orbitals of Cbz-PrA⁻ and PTC4A⁴⁻ ligands, which corresponding to $L_{(Cbz-PrA+PTC4A)}CT$ transition.” (Page 4)*

5. The description of Cu12b is similar to a class of solid materials for adsorption and separation the nonporous adaptive crystals (NACs), which function at the supramolecular level (*J. Am. Chem. Soc.* 2022, 144, 113–117; *Chem* 2023, 9, 1-9), thus, this kind of materials

can be mentioned.

Response: Thanks for your kind reminder of the reviewer. Compared with traditional porous materials, such as supramolecular organic frameworks, metal-organic frameworks (MOFs), hydrogen-bonded frameworks (HOF), the newly emerged nonporous adaptive crystals (NACs) usually possess large intrinsic surface area and well-defined pore structure under certain guest molecules stimulation (*J. Am. Chem. Soc.* 2022, 144, 113–117; *J. Am. Chem. Soc.* 2021, 143, 18849–18853; *Angew. Chem. Int. Ed.* 2021, 60, 1690 – 1701; *iScience* 2020, 23, 101443). In this work, the post-transformation product Cu12b possesses variation of porosity before and after the activation, as well as an intrinsically large surface area, which can be viewed as NACs for further investigation. Therefore, we redefine the guest-free Cu12b as a potential nonporous adaptive crystal in accordance with its adaptive properties, and relevant references have been cited in the revised manuscript yet.

New comments and references have been added in the revised manuscript as: “thus the Cu12b-NACs can be viewed as cluster-based nonporous adaptive crystals (NACs), which are relatively nonporous but with medium surface areas.” (Page 11)

“55. Li, B., Li, S., Wang, B., Meng, Z., Wang, Y., Meng, Q. & Li, C. Capture of Sulfur Mustard by Pillar[5]arene: From Host-Guest Complexation to Efficient Adsorption Using Nonporous Adaptive Crystals. *iScience* 23, 101443, (2020).

56. Luo, D., He, Y., Tian, J., Sessler, J. L. & Chi, X. Reversible Iodine Capture by Nonporous Adaptive Crystals of a Bipyridine Cage. *J. Am. Chem. Soc.* 144, 113-117 (2022).

57. Luo, D., Tian, J., Sessler, J. L. & Chi, X. Nonporous Adaptive Calix[4]pyrrole Crystals for Polar Compound Separations. *J. Am. Chem. Soc.* 143, 18849-18853 (2021).

58. Sheng, X. Li, E., Zhou, Y., Zhao, R., Zhu, W. & Huang, F. Separation of 2-Chloropyridine/3-Chloropyridine by Nonporous Adaptive Crystals of Pillararenes with Different Substituents and Cavity Sizes. *J. Am. Chem. Soc.* 142, 6360-6364 (2020).

59. Wu, J. R. & Yang, Y. W. Synthetic Macrocyclic-Based Nonporous Adaptive Crystals for Molecular Separation. *Angew. Chem., Int. Ed.* 60, 1690-1701 (2021).

60. Zhou, J., Yu, G., Li, Q., Wang, M. & Huang, F. Separation of Benzene and Cyclohexane by Nonporous Adaptive Crystals of a Hybrid[3]arene. *J. Am. Chem. Soc.* **142**, 2228-2232 (2020)."

(Page 19)

6. How to prepare the polyiodide anions in the solid state to test UV-Vis absorption? And why the UV-Vis absorption spectra of Cu12a-2D π OF and the polyiodide anions show a similar absorption peak at 300-500 nm?

Response: Thanks for your constructive suggestions of the reviewer. Initially, we assumed that polyiodide anion (mainly I₃⁻ and I₅⁻) presented as the iodine species in I₂@Cu12a- π , thus we physically mixed pristine iodine and KI, and ground them thoroughly to obtain the polyiodide anions, which seems to be less rigorous to obtain in this way. Thus, we have corrected the polyiodide anions into the pristine iodine in the UV-Vis absorption and compared the differences of them (*Macromolecules* 2016, 49, 6322-6333; *Angew. Chem. Int. Ed.* 2022, 61, e202113724).

The comments have been corrected in the revised manuscript as: "*The UV-Vis absorption spectra of Cu12a- π and the pristine iodine showcased a significantly different absorption profile compared to I₂@Cu12a- π (Fig. 6e) in the solid state. The Cu12a- π and the pristine iodine exhibited weak absorption intensity beyond 900 nm, yet the latter exhibited a broad tail peak over 1000 nm.*"

(Page 13)

Fig. 6 **a** Time-dependent adsorption profiles for the **Cu12a- π** and **Cu12b-NACs** (0.1 mg mL^{-1}) when immersed in 30 mL iodine aqueous solutions (1.2 mM). **b** UV-Vis spectra of saturated iodine aqueous solutions before and after adsorption by **Cu12a- π** and **Cu12b-NACs**, respectively; the inset shows the color changes of the iodine aqueous solution after adsorption by **Cu12a- π** and **Cu12b-NACs**, respectively. **c** SEM image and EDS of **I₂@Cu12a- π** , scan bar= $100 \mu\text{m}$. **d** PXRD patterns of **Cu12a- π** , **Cu12a- π in H₂O** and **I₂@Cu12a- π** . **e** UV-Vis spectra of **Cu12a- π** , **I₂@Cu12a- π** and **I₂**. The inset shows color change of the **Cu12a- π** before and after adsorbing iodine. **f** XPS spectrum of **I₂@Cu12a- π** .

7. Except for the cavity of this crystal contributes to the absorption of I₂, are there any other interaction that make **Cu12a** exhibits high iodine adsorption capacity? Will I₂ react with Cu?

Response: Thank you for these insightful comments. We understand the reviewers' concern here. Indeed, in the part of iodine adsorption, we have carried out the iodine adsorption measurement for both **Cu12a- π** and **Cu12b-NACs**. In comparison with the **Cu12a- π** with pore structure, **Cu12b-NACs** are relatively nonporous but still possess considerable iodine adsorption capacity up to 2.73 g g^{-1} . Consequently, we deduced that iodine adsorption sites of the **Cu12a- π** and **Cu12b-NACs** are not only the cavity of the crystal, but also related to the surface areas. Moreover, since the **Cu₁₂** metal kernel is completely capped by the peripheral ligands, the I₂ cannot react with Cu during the adsorption process.

The comments have been corrected in the revised manuscript as: "Therefore, we can draw a preliminary conclusion that the iodine adsorption sites of the **Cu12a- π** and **Cu12b-NACs** are not only the cavity of the supramolecular framework, but also are related to the surface areas of crystal."

(Page 13)

8. There are some details need to be proved. The format of caption needs to be the same. In supporting information, the figure in "It was apparent that both **Cu12a** and **Cu12b** featured similar electronic structures (Supplementary Fig. 16)" should be Supplementary Fig. 17.

Response: Thanks for the kind reminder of the reviewer. We have carefully checked the

full manuscript and corrected the format of caption, and the Supplementary Fig. 16 has been corrected.

We will appreciate it if the reviewers can understand our endeavor! Thank you very much for your insightful suggestion!

Reviewers' Comments:

Reviewer #1:

Remarks to the Author:

My concerns have been properly addressed. I recommend to accept the manuscript!

Reviewer #2:

Remarks to the Author:

In my original report, I commented about the potential novelty and impact of the results, which I considered limited based on the reported data. In this revised version, the authors have made additional changes in response. I appreciate their effort to address the points raised by this and the other reviewers. However, I don't believe that the revisions provide any significant addition in these respects, so my opinion remains that this manuscript is more suitable for a more specialized journal. Changes have been made to the employed language and nomenclature, to make it more consistent with the results, and the existing literature, especially for supramolecular networks. Otherwise, the new experimental data regarding the separation properties, such as the breakthrough experiments, prove that the new compounds do not show any significant potential for this application. Regarding the interpretation of the gas isotherms, and its relation to the compounds flexibility, there are still errors in this revised version. The authors insist on the presence of mesopores, but there is no hysteresis in the nitrogen sorption isotherm. The uptake for the recorded adsorption and desorption points all coincide. In addition, the pore size values obtained from the HK model do not belong to the mesopore regime. Therefore, the authors' claims about flexibility or adaptability are not experimentally supported.

Reviewer #3:

Remarks to the Author:

In this revised manuscript, Sun and coworkers conducted additional experiments that comprehensively address all of my concerns. Furthermore, the quality of the manuscript has been significantly enhanced. Overall, both the experimental work and the novelty of the research meet the high standards of Nature Communications. Therefore, I highly recommend its publication without any further revisions.

Responses to the Reviewers' Comments and the Corresponding Revisions

Reviewer #1 (Remarks to the Author):

My concerns have been properly addressed. I recommend to accept the manuscript!

Response: Thanks for reviewer's comments. According to reviewer's comments, we have carefully revised and supplemented the manuscript, which greatly improved the quality of our research work. Thanks for the reviewer again.

Reviewer #2 (Remarks to the Author):

In my original report, I commented about the potential novelty and impact of the results, which I considered limited based on the reported data. In this revised version, the authors have made additional changes in response. I appreciate their effort to address the points raised by this and the other reviewers. However, I don't believe that the revisions provide any significant addition in these respects, so my opinion remains that this manuscript is more suitable for a more specialized journal.

Response: According to the previous comments and suggestions of the reviewer, we have carefully revised the manuscript, which undoubtedly brought great help for the improvement of our work. Therefore, thanks for the reviewer's valuable comments and suggestions on our work.

As we have replied to the reviewer last time, we believe our work not only gives important findings in macrocycle/cluster-based porous supramolecular frameworks (PSFs), but also inspires researchers working in inorganic chemistry, cluster chemistry, nanochemistry and so on. Since the reviewer still has doubts on the innovation on our work, we would like to emphasize the significance and innovation on our work in detail again. We do think our innovations and experiments provided enlightenment to the following crucial questions:

1. How to address the issue of insufficient structural stability in traditional organic π -stacked organic frameworks (π OFs)?

Organic molecules-based building unit as a typical representative has been recognized in π -stacked porous materials, such as π OFs, π -stacked polymer and so on (*PNAS* 117, 20397 (2020); *CCS Chem.* 4, 1315–1325 (2022); *Angew. Chem. Int. Ed.* 61, e202201646 (2022); *Polymer* 144, 51-56, (2018)). However, the construction of π -stacked porous framework with robustness is challenging, because the framework constructed by noncovalent $\pi \cdots \pi$ /C-H $\cdots \pi$ interactions (weak, flexible and poor directionality) will most likely collapse after solvent removal/re-adsorption and de-compose upon heating. Although several works on the organic molecules-based π OFs have been reported, few of the frameworks could maintain their stability especially under high temperature (*PNAS* 117, 20397 (2020)). Nevertheless, metal nanocluster as building

units can be applied in the construction of cluster-based metal organic frameworks (cluster-based MOFs), which have shown great application in gas separation, catalysis, detection and so on, and its framework showcases great structural stability due to the robust cluster-based building units (*Nature Chem.* 9, 689–697 (2017); *Angew. Chem. Int. Ed.* 59, 2-8 (2020)). Inspired by above, in our work, we combine the weak/flexible C-H \cdot π interactions and the robust metal nanocluster to obtain a dynamic and robust π -stacked porous supramolecular framework (**Cu12a- π**). Benefiting from the former, the framework of such PSF can achieve reversible SC-SC transformation by removal and re-adsorption of the guest molecule, which demonstrates excellent dynamics of the framework. As for the latter, the inherent robust cluster-based building unit (Cu₁₂) endows the PSF with superior thermo-stability, showing no change in variable-temperature PXRD patterns even up to 513 K. This π -stacked PSF not only solves the poor stability of organic molecule-based π OFs but also successfully turns this defect into an advantage in terms of framework dynamic.

2. Does the dynamic coordination bonds of metal clusters offer innovative avenues for designing and synthesizing PSFs?

As the review's suggestion last time, we have corrected the "post-modification" into the "transformation". "Transformation" is also a new pathway to re-construct PSFs, especially in metal cluster-based PSFs. This arises from the inherent dynamic nature of coordination bonds in solution, which offers a chance to alter their structures in some case (*Chem. Rev.* 111, 6810–6918 (2011); *Proc. Natl Acad. Sci. USA* 114, 12132–12137 (2017)). When this kinds of metal clusters act as building unit in PSFs, the dynamic characteristics of the coordination bonds will enable the PSF to transform in response to some external stimulus, so as to re-build the PSF. In our work, due to the dynamic characteristic of Cu-S, Cu-O bonds, the **Cu12a- π** can be transformed into the **Cu12b-NACs** under the stimulation of *p-phenyl*-thiacalix[4]arene, leading to the reconstruction of supramolecular framework. This kinds of transformation approach based on dynamic characteristics of coordination bonds in metal cluster will give more structural transformation approaches and possibilities of cluster-based porous materials, for instance cluster-based MOFs and PSFs.

In addition to the previous contents of our work, we also performed an in-depth investigation into the transformation process and underlying mechanism to further improve the integrity and quality of our work. Electrospray ionization mass spectrometry (ESI-MS), a widely recognized and valuable analytical technique, has been utilized to track the transformation process and yield vital speciation information. In the revised stage, we carried out ESI-MS measurement for **Cu12a**, **Cu12b** and their ligand-exchange process (**Cu12a** to **Cu12b**). As presented in Fig. 5c-d, both **Cu12a** and **Cu12b** show parent peaks in positive-ion mode ESI-MS (**1a-1e** for **Cu12a**; **2a-2i** for **Cu12b**), suggesting their outstanding solution stability. We further tracked the solution species evolution over the course of 80 min upon addition of macrocycle ligand (H₄PTC4A) in the CHCl₃ solution of **Cu12a** cluster by time-dependent ESI-MS (Fig. 5e). Of note, we not only detected the final transformed product, **Cu12b** (**2j**), after 80 minutes of reaction, but also successfully identified a crucial intermediate species, **3a**. This intermediate corresponds to the mono-substituted product of **Cu12a**, where one of the TC4A⁴⁺ ligands in **Cu12a** is replaced by PTC4A⁴⁺.

Fig. 5 a Structural transformation of monomer Cu₁₂ cluster from Cu12a to Cu12b. b UV-Vis absorption spectra of the conversion from Cu12a to Cu12b over the time course at 293 K. c, d

Positive-ion mode ESI-MS of the crystals of **Cu12a** (a) and **Cu12b** (d) dissolved in mixed CHCl_3 and CH_3OH . Inset: The zoom-in mass spectrum of experimental (green line) and simulated (red line) isotope-distribution patterns of each labeled species. (e) Time-course ESI-MS of transformation from **Cu12a** to **Cu12b** induced by adding $\text{H}_4\text{PTC4A}$.

Supplementary Fig. 22 (a) Collision-induced dissociation (CID) mass spectra of **Cu12a** measured at the collision energy of 0–100 eV. (b) Comparison of the experimental (green line) and simulated (red line) isotopic patterns of **1f**. (c) Relative intensity of the resulting **1b** and **1f** collected at the collision energy of 0–100 eV. (d) CID mass spectra of **Cu12b** measured at the collision energy of 0–100 eV. (e) Relative intensity of the resulting **2b** collected at the collision energy of 0–100 eV.

Besides, we also performed the collision-induced dissociation (CID) mass to compare the gas-phase stability between the two building units. As shown in Supplementary Fig. 22, by increasing the collision energy from 0 to 100 eV, the species of **Cu12a** progressively

faded away. While a new +1 charged species **1f** was observed as the energy increased, corresponding to the dissociation of Cbz-PrA⁻ ligand. However, this phenomenon was not observed in **Cu12b** and no fragments were generated through the increasing of collision energies under the same conditions. Such results indicate that the **Cu12b** possesses higher gas-phase stability than that of **Cu12a**.

Based on above experiments, we infer that the transformation process involves a "dissociation-reassembly (DR)" mechanism, in which the process undergoes a kinetically driven dissociation of the initial cluster and then reassemble into a thermodynamically stable new cluster. More importantly, in comparison with the assumed mechanism based on the single crystal structure and optical monitoring (simple ligand-exchange process), such DR mechanism provided by ESI-MS given more in-depth and accurate conceptual framework for understanding the intricate process involved in the transformation of **Cu12a** to the stable **Cu12b** cluster.

Changes have been made to the employed language and nomenclature, to make it more consistent with the results, and the existing literature, especially for supramolecular networks. Otherwise, the new experimental data regarding the separation properties, such as the breakthrough experiments, prove that the new compounds do not show any significant potential for this application.

Response: Thank you for the constructive suggestion you provided regarding the "two-dimensional" and "organic frameworks" during the previous review, which greatly helps us to correct our understanding and investigation of such π -stacked PSFs. Although the **Cu12a- π** did not show potential gas separation performance in the breakthrough experiments, yet we have retained the basic gas adsorption contents in the revised manuscript to confirm the real existence of permanent porosity in **Cu12a- π** . Furthermore, the highlights of this work are mainly the dynamic and transformable π -stacked framework and the considerable pristine iodine adsorption in aqueous solution, rather than gas separation, and thus we believe that the poor gas separation performance does not lower the integrity and innovation of this work. In order to compensate the limitations of the low gas adsorption and separation performances, we further conducted a large number of ESI-MS experiments on the transformation part, one of the highlights of this

work, to reveal the vital speciation information in the transformation process (as discussed in last question). This not only enriches the contents of this work, but also makes it more readable and innovative.

Regarding the interpretation of the gas isotherms, and its relation to the compounds flexibility, there are still errors in this revised version. The authors insist on the presence of mesopores, but there is no hysteresis in the nitrogen sorption isotherm. The uptake for the recorded adsorption and desorption points all coincide. In addition, the pore size values obtained from the HK model do not belong to the mesopore regime. Therefore, the authors' claims about flexibility or adaptability are not experimentally supported.

Response: As the interpretation of the gas isotherms in the last review stage, we mistakenly assumed that the slightly smaller difference between N₂ adsorption and desorption isotherms attributes to the hysteresis, which mislead our interpretation of N₂ isotherms. Meanwhile, the corresponding pore size distribution is not in the mesoporous range. Therefore, we have re-interpreted the gas sorption isotherms to confirm the real existence of permanent porosity in **Cu12a-π**.

The corresponding description has been corrected in paragraph 2 on page 7 in the revised manuscript (highlighted in red).

"Correspondingly, analysis of N₂ adsorption isotherms of Cu12a-π gives rise to a Brunauer-Emmett-Teller (Langmuir) surface area of 31.93 (45.29) m²·g⁻¹, and a wide pore size distribution in the range of 0.8–1.6 nm calculated by Horvath-Kawazoe (HK) model (Fig. 4e and Supplementary Fig. 9). These results are thought to reflect the small cavities provided by the thiocalix[4]arene and the voids between the 2D layers. Meanwhile, Fig. 4f shows the pure-component equilibrium adsorption isotherms for the CH₄, CO₂, C₂H₄ and C₂H₆ collected at different temperatures. The maximum C₂H₆ uptake reaches 30.48 cm³·g⁻¹ (273 K) at normal pressure, further confirming the real existence of permanent porosity in Cu12a-π."

In summary, we believe that our work is comprehensive, innovative and will be definitely helpful not only to researchers in porous chemistry, but also in inorganic chemistry, cluster chemistry, nanochemistry and so on. We will appreciate it if the reviewers can understand our endeavor! Thanks for the review's comments again.

Reviewer #3 (Remarks to the Author):

In this revised manuscript, Sun and coworkers conducted additional experiments that comprehensively address all of my concerns. Furthermore, the quality of the manuscript has been significantly enhanced. Overall, both the experimental work and the novelty of the research meet the high standards of Nature Communications. Therefore, I highly recommend its publication without any further revisions.

Response: We greatly appreciate the reviewer's professional and detailed comments and suggestions, which have been instrumental in improving the quality of our research. As a result of the reviewer's feedback, our manuscript has undergone significant improvements since its original submission. We express our sincere gratitude to the reviewer for their valuable contribution.

Reviewers' Comments:

Reviewer #2:

Remarks to the Author:

The authors have corrected the gas sorption data misinterpretation and removed comments about hysteresis and mesoporosity. They have also added a mass spectroscopy study to monitor the transformation from compounds Cu12a to Cu12b. The results indicate cluster dissociation before formation of final Cu12b product, and based on the identification of partially substituted species, the authors propose a dissociation-reassembly mechanism. With this, they suggest that compound 12b is thermodynamically more stable, although this is not supported by any energy calculation.

At this point, I have no further comments on data interpretation or experimental design. I appreciate the authors' efforts to convince me of the significance of their work in the rebuttal letter. However, my opinion on novelty and potential impact remains the same as in previous reports, as the core results and scientific content of the manuscript have remained fundamentally the same after the review process.

Responses to the Reviewers' Comments and the Corresponding Revisions

Reviewer #2 (Remarks to the Author):

The authors have corrected the gas sorption data misinterpretation and removed comments about hysteresis and mesoporosity. They have also added a mass spectroscopy study to monitor the transformation from compounds Cu12a to Cu12b. The results indicate cluster dissociation before formation of final Cu12b product, and based on the identification of partially substituted species, the authors propose a dissociation-reassembly mechanism. With this, they suggest that compound Cu12b is thermodynamically more stable, although this is not supported by any energy calculation. At this point, I have no further comments on data interpretation or experimental design. I appreciate the authors' efforts to convince me of the significance of their work in the rebuttal letter. However, my opinion on novelty and potential impact remains the same as in previous reports, as the core results and scientific content of the manuscript have remained fundamentally the same after the review process.

Response: Thanks for reviewer's constructive comments and for the recognition of our revised contents last time. According to your comments, we have carefully revised and supplemented the manuscript through solid and systematic works, which have greatly improved the quality of our research work. Meanwhile, we agree that the proposal on the energy calculation of **Cu12a** and **Cu12b** is helpful and meaningful, which will be able to reveal the mechanism of transformation by combining experimental results.

To compare the energies of these two Cu₁₂ clusters via DFT calculation, it is essential to ensure the conservation of their chemical formulas, or the result will be meaningless. However, the formulas of **Cu12a** and **Cu12b** are not identical and therefore their energies from DFT calculations cannot be directly compared. In order to overcome the defect of DFT calculation and indirectly compare the energy difference between the two clusters, we maintained the conservation of materials before and after the conversion reaction ($\text{Cu12a} + 2 \times \text{PTC4A} \rightarrow \text{Cu12b} + 2 \times \text{TC4A}$), and then calculated the energies using DFT. Our calculations indicate that the transformation from **Cu12a** to **Cu12b** is an exothermic reaction ($\Delta E = 48.16$ eV: the overestimation of such energy attributes to the breakage and

formation of coordination bonds in above computational system), which qualitatively reflects that Cu12b is more thermodynamically stable.

New comments have been added in the revised manuscript and Supplementary Figures as: “This result indicates that Cu12b is more thermodynamically stable than Cu12a, which also was confirmed by DFT calculations (Supplementary Fig. 11).” (Page 10)

Supplementary Fig. 11 The relative energy difference between the initial and final stages of the transformation reaction calculated by Gaussian 16 (PBE1PBE level).

In summary, we believe that our work is comprehensive, innovative and will be definitely helpful not only to researchers in porous chemistry, but also in inorganic chemistry, cluster chemistry, nanochemistry and so on. We will appreciate it if the reviewers can understand our endeavor! Thanks for the review's comments again.

Reviewers' Comments:

Reviewer #2:

Remarks to the Author:

The calculated energy values for the exchange reaction are in agreement with the claims for thermodynamic stability. The value given in the SI seems reasonable, although this is different to the one in the response letter, which appears to be overestimated, as explained by the authors. In sum, there is no further technical comment from my side, and the paper is thus ready for publication.

Reviewer #4:

Remarks to the Author:

The Editor asked me to evaluate the validity of quantum-chemical simulations of energy difference between the initial and final stages of the transformation reaction Cu12a → Cu12b supporting experiment. Upon inspection of the revised MS, SI and the response letter, I find the following:

1. RESPONSE LETTER: The figure given in the response letter (Supplementary Fig. 11) is incorrect and is different from the actual Supplementary Fig. 11. The relative energy difference of 48 eV in this 'incorrect' figure appears to be questionable; the magnitude of this value seems excessively high to be considered valid. Furthermore, the description of the simulation methodology provided in the response letter is unclear and may benefit from additional clarification.

2. SUPPLEMENTARY INFORMATION, Figure 11. The energy difference presented in this context, which is 7.32 kcal/mol, represents the Gibbs free energy difference. This is the appropriate measure for drawing conclusions regarding the direction of a chemical reaction. Furthermore, the reported value of 7.32 kcal/mol aligns with our physical intuition and corroborates the thermodynamic stability of Cu12b, consistent with the experimentally observed direction of the chemical reaction.

3. SUPPLEMENTARY INFORMATION. Subsection "Energies calculation details" on page 4: The paragraph briefly introduces the calculations of Gibbs free energy. However, I believe that this description lacks detail, and it is important to provide more information to help the reader understand the methodology employed in these simulations. Starting from the general definition of $\Delta G = \Delta E + \Delta ZPE$, it would be beneficial to provide a comprehensive explanation of each component and the specific methods used to obtain ΔE and ΔZPE . Furthermore, I recommend including a supplemental table within this subsection, which would present a tabulated summary of the respective ΔE and ΔZPE values for all four compounds under consideration, namely Cu12a, Cu12b, PTC4A, and TC4A. This additional table will enhance the clarity of the presented data and facilitate a better understanding of the results. Finally, Geometries used for calculations of these quantities should be uploaded as a part of SI.

4. MAIN TEXT: I believe that the article contains an excessive number of acronyms, including even the abstract. I recommend that the authors make an effort to reduce the use of acronyms in order to enhance the readability of the article for a wider readership, especially considering the broad audience of Nature Communications.

5. MAIN TEXT: Fig. 1 Description of cluster-based PSF. I believe that this figure offers a visually informative representation. However, the authors could enhance the text by referencing atomistic images of the molecular building blocks already presented in the Supplementary Information. This would provide readers with a clearer perception of the structures involved. Additionally, the term 'guest' is not well-defined in this context, which may require readers to delve deeper into the paper to comprehend the chemical structures being referred to.

Responses to the Reviewers' Comments and the Corresponding Revisions

Reviewer #2 (Remarks to the Author):

The calculated energy values for the exchange reaction are in agreement with the claims for thermodynamic stability. The value given in the SI seems reasonable, although this is different to the one in the response letter, which appears to be overestimated, as explained by the authors. In sum, there is no further technical comment from my side, and the paper is thus ready for publication.

Response: Thanks for reviewer's comments. According to reviewer's comments, we have carefully revised and supplemented the manuscript, which greatly improved the quality of our research work. Thanks for the reviewer again.

Reviewer #4 (Remarks to the Author):

The Editor asked me to evaluate the validity of quantum-chemical simulations of energy difference between the initial and final stages of the transformation reaction Cu₁₂a → Cu₁₂b supporting experiment. Upon inspection of the revised MS, SI and the response letter, I find the following:

We greatly appreciate the reviewer's professional and detailed comments, which have been instrumental in improving the quality of our research.

1. RESPONSE LETTER: The figure given in the response letter (Supplementary Fig. 11) is incorrect and is different from the actual Supplementary Fig. 11. The relative energy difference of 48 eV in this 'incorrect' figure appears to be questionable; the magnitude of this value seems excessively high to be considered valid. Furthermore, the description of the simulation methodology provided in the response letter is unclear and may benefit from additional clarification.

Response: Thanks for your professional suggestion. In the previous revision, we accepted the reviewer's comments and established a relatively ideal conversion reaction (Cu₁₂a+2×PTC4A→Cu₁₂b+2×TC4A), and calculated their energy difference following the equation below:

$$\Delta E_{[\text{Cu}_{12}\text{a}+2\times(\text{PTC4A})\rightarrow\text{Cu}_{12}\text{b}+2\times(\text{TC4A})]} = [E_{(\text{Cu}_{12}\text{a})}+2\times E_{(\text{PTC4A})}] - [E_{(\text{Cu}_{12}\text{b})}+2\times E_{(\text{TC4A})}]$$

However, such equation suffers from the interference of the calculation method, i.e., different basis sets are used for the two components of clusters and organic ligands (clusters used pseudopotential and conventional basis sets: LanL2DZ+6-31G*; the organic ligand only used the conventional basis sets: 6-31G*), which leads to an excessive energy difference in the calculation ($\Delta E=48.16$ eV).

To solve this drawback, we further optimize our computational model below:

$$\Delta G_{[\text{Cu}_{12}\text{a}+2\times(\text{PTC4A})\rightarrow\text{Cu}_{12}\text{b}+2\times(\text{TC4A})]} = E_{[\text{Cu}_{12}\text{a}+2\times(\text{PTC4A})]} - E_{[\text{Cu}_{12}\text{b}+2\times(\text{TC4A})]} + \Delta ZPE.$$

This model combines the cluster and introduced ligands in one computational input file to calculate the energies, giving a relatively reasonable energy difference of 7.32 kcal/mol. This result reflects that the conversion reaction is an exothermic reaction, in which the reaction proceeds towards a more stable product.

Finally, the description of the calculation methodology have been further supplemented in the revised Supplementary Information.

2. SUPPLEMENTARY INFORMATION, Figure 11. The energy difference presented in this context, which is 7.32 kcal/mol, represents the Gibbs free energy difference. This is the appropriate measure for drawing conclusions regarding the direction of a chemical reaction. Furthermore, the reported value of 7.32 kcal/mol aligns with our physical intuition and corroborates the thermodynamic stability of Cu12b, consistent with the experimentally observed direction of the chemical reaction.

Response: Thanks for your professional suggestion and recognition. Although a cluster transformation reaction involves many complex factors, we still hope that the established transformation model and corresponding energy calculations can effectively indicate the direction of the chemical reaction as much as possible.

3. SUPPLEMENTARY INFORMATION. Subsection “Energies calculation details” on page 4: The paragraph briefly introduces the calculations of Gibbs free energy. However, I believe that this description lacks detail, and it is important to provide more information to help the reader understand the methodology employed in these simulations. Starting from the general definition of $\Delta G = \Delta E + \Delta ZPE$, it would be beneficial to provide a comprehensive explanation of each component and the specific methods used to obtain ΔE and ΔZPE . Furthermore, I recommend including a supplemental table within this subsection, which would present a tabulated summary of the respective ΔE and ΔZPE values for all four compounds under consideration, namely Cu12a, Cu12b, PTC4A, and TC4A. This additional table will enhance the clarity of the presented data and facilitate a better understanding of the results. Finally, Geometries used for calculations of these quantities should be uploaded as a part of SI.

Response: Thanks for your constructive suggestion. According to your suggestions, we have added the detail methodology of energy calculations in the Subsection “Energies calculation details”. As mentioned above, to minimize the calculation error, we combine the clusters (Cu12a or Cu12b) and introduced ligands (TC4A or PTC4A) into one computational input file (i.e., [Cu12a + 2(PTC4A)].gjf or [Cu12b + 2(TC4A)].gjf) to calculate their electronic energy (E), the zero-point energy (ZPE) obtained from

vibrational frequency analysis was used to correct the electronic energy (E). Thus, the Gibbs free energies for the transformation was calculated as follow: $\Delta G = G_{[\text{Cu12a} + 2(\text{PTC4A})]} - G_{[\text{Cu12b} + 2(\text{TC4A})]} = E_{[\text{Cu12a} + 2(\text{PTC4A})]} - E_{[\text{Cu12b} + 2(\text{TC4A})]} + \Delta ZPE = 7.32 \text{ kcal/mol}$.

The additional table containing the energies of each species, and geometries of the calculated file have been added in the revised Supplementary Information.

Species	E _{ele}	ZPE
[Cu12a + 2(PTC4A)]	-19235.570803 Ha	3.563644 Ha
[Cu12b + 2(TC4A)]	-19235.590445 Ha	3.571621 Ha

1 Ha = 627.5094 kcal/mol.

4. MAIN TEXT: I believe that the article contains an excessive number of acronyms, including even the abstract. I recommend that the authors make an effort to reduce the use of acronyms in order to enhance the readability of the article for a wider readership, especially considering the broad audience of Nature Communications.

Response: Thank you for pointing this out! We have reduced the acronyms in the revised manuscripts, particularly in the abstract, to enhance readability for readers.

5. MAIN TEXT: Fig. 1 Description of cluster-based PSF. I believe that this figure offers a visually informative representation. However, the authors could enhance the text by referencing atomistic images of the molecular building blocks already presented in the Supplementary Information. This would provide readers with a clearer perception of the structures involved. Additionally, the term 'guest' is not well-defined in this context, which may require readers to delve deeper into the paper to comprehend the chemical structures being referred to.

Response: Thanks for reviewer's constructive and kind comments regarding Fig. 1. Based on your suggestions, we have added an exact image of the building block of **Cu12b** to Fig. 1 and included a detailed description of the guest in the caption, aiming to provide a clear summary of the text through Fig. 1 to readers.

The revised Fig. 1 and corresponding caption have been corrected on page 3 in the revised manuscript (highlighted in red).

“Fig. 1 Description of cluster-based PSF. Cartoon illustration showing the construction of cluster-based building unit. The Cu_{12} cluster is further assembled into PSF via C-H $\cdots\pi$ interactions; NACs represent the nonporous adaptive crystals; the guest in $Cu_{12}a$ is ethyl acetate (EA) molecule.”

In summary, we are very grateful to the reviewers for your constructive suggestions, which have helped us improve the quality of this work. We will appreciate it if the reviewers can understand our endeavor! Thanks for the review’s comments again.

Reviewers' Comments:

Reviewer #4:

Remarks to the Author:

The authors have adequately responded to the referee's suggestions and appropriately revised the paper. The revision benefitted the MS. I recommend the article to be accepted in Nature Comm for publication.